# DenseAttention: No-Compromise Exact All $N \times N$ Interactions Algorithm with $O(N)$ Space and Time Complexity

## Abstract

The ubiquitous Transformer architecture suffers from two main bottlenecks: 1) low computational and memory efficiency, leading to suboptimal hardware utilization, and 2) quadratic time complexity with respect to sequence length $N$, making it slow and costly for large data contexts. We propose a novel DenseAttention Network architecture, a straightforward simplification of the standard Transformer block that addresses these issues and serves as a drop-in replacement for language modeling tasks. We eliminate memory-bound components in DenseAttention, including Softmax, masking, one skip connection, and both LayerNorms, as well as key, value, and output projection matrices, as they become redundant. Despite these removals, it maintains exact $N \times N$ pairwise interactions between tokens. By exploiting the associativity of matrix multiplications, DenseAttention can be computed with $O(N^2 d)$ or $O(N d^2)$ time and space complexity, depending on the context. To handle the absence of Softmax and prevent numerical instability, we introduce MaxNormActivation at both ends of the Transformer block. We also devise Cosine Relative Positional Embeddings as a computationally efficient replacement for RoPE, and simple LocalAttention variations of the block to help the model focus on details in extremely long contexts.

DenseAttention competes with FlashAttention in speed on small sequences and outperforms it by orders of magnitude on large contexts. We pre-train encoder language models on sequences up to 16K in length, which perform similarly or better than baseline BERT-large, while significantly improving speed and efficiency. Finally, we achieve state-of-the-art on the LRA benchmark among the Transformer-based architectures.

## 1 Introduction

Transformer architecture (Vaswani et al., 2017) has become ubiquitous in neural networks across many domains and modalities, such as NLP (Devlin et al., 2019), images (Dosovitskiy et al., 2021), video (Arnab et al., 2021), speech recognition (Radford et al., 2022), and even tabular data (Arik & Pfister, 2019)). But most notably, it's the core component of Large Language (Touvron et al., 2023a; Brown et al., 2020) and Multi-modal (Bai et al., 2023) Models, which demonstrate surprisingly good abilities in natural language understanding, comprehension and reasoning tasks.

The most prominent feature which distinguishes a Transformer layer from other architectures is the attention mechanism which allows for all of the inputs to simultaneously interact with each other. However, it's also the source of its limitations: $O(N^2)$ time and space complexity w.r.t context length $N$, and computational inefficiency of the constituents which make the architecture work seamlessly. As reported by Ivanov et al. (2021), matrix multiplications account for 99.8% of total FLOPs during BERT pretraining and only 61% of runtime, the discrepancy being caused by low arithmetic intensity of memory bound operations, namely, LayerNorms, Softmaxs and other activations as well as elementwise operations.

Numerous extensions and modifications to the standard Transformer (Katharopoulos et al., 2020; Choromanski et al., 2022; Beltagy et al., 2020; Zhai et al., 2021; Hua et al., 2022) have been proposed in the recent years to alleviate the restrictive $O(N^2)$ complexity. However, as these architec-

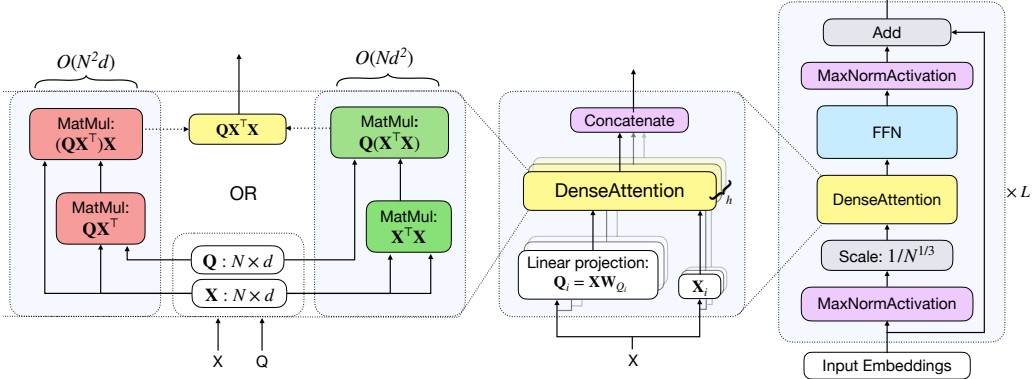

Figure 1: DenseAttention architecture. Left: DenseAttention mechanism; center: multi-head interpretation; right: the entire DenseAttention Network

tures in general rely on non-linear, memory-intensive and sparse operations to a much greater degree than traditional attention mechanism, their throughput in terms of tokens per second and hardware utilization are subpar in comparison with the latter on all but large sequence lengths (Tay et al., 2022; Dao et al., 2022). Besides, some report (Xiong et al., 2022; Sun et al., 2024; Tay et al., 2023), that their modeling capabilities may be limited in comparison with full-rank exact attention while their conceptual complexity and incompatibility with standard architectures prevents their widespread adoption.

Thus, we aim to achieve 3 main goals:

1. To create hardware efficient yet hardware-agnostic architecture with the arithmetic intensity ratio as high as possible. An ideal algorithm should contain merely matrix multiplications with no activations, normalizations and residual connections. However, while possible in principle, it remains a challenging task due to numerical instabilities occurring both in forward and backward pass and lagging performance of such architectures (Balduzzi et al., 2017; Santurkar et al., 2018; Pascanu et al., 2013)

2. To create an algorithm which would efficiently process long sequences, preferrably with $O(N)$ time and space complexity.

3. To make the resulting architecture as simple as possible, and closely resembling original Transformer architecture as well so it can serve as a drop-in replacement for the former and be easily adopted by both research and practitioners communities.

We accomplished all of these goals with DenseAttention and DenseAttention Network (DANet) blocks (Fig. 1). This architecture is a straight-forward simplification of the traditional Transformer architecture which does not introduce any additional elements and complexities to the module and can be freely swapped with it. On the contrary, we develop DenseAttention by *removing* all computationally inefficient elements of the original architecture: biases in all linear layers, masks, dropout, residual connection between attention and FFN. Most importantly, we remove Softmax inside self-attention. It results in the whole scaled dot-product attention mechanism becoming just a composition of matrix multiplications, which can be done in any order by associative property of matrix multiplication. This duality allows to calculate DenseAttention using either $O(N^2 d)$ or $O(N d^2)$ FLOPs, and the second option has linear time and space complexity w.r.t sequence length.

We remove LayerNorms and instead use a new MaxNormActivation, which scales token representations by their $l_\infty$ norm. We place it at both ends of the DANet block. We also remove all projection matrices except $W_Q$ in the self-attention module as they become redundant in the absence of non-linearities between attention and FFN. To empirically validate the architecture, we test on the challenging Long Range Arena (LRA) benchmark (Tay et al., 2021) and achieve a new SOTA result across all of the transformer-based models, even competing with State-Space-Models (Gu et al., 2022a). We also replace Transformer modules in BERT-large model (Devlin et al., 2019) with DenseAttention Network modules and pre-train it from scratch on sequences up to 16k tokens.

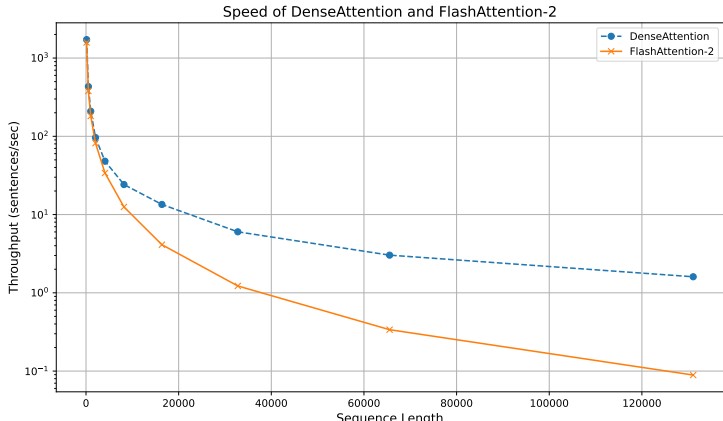

Figure 2: Comparison of speed between DenseAttention and FlashAttention2 (Dao (2024)) models across sequence lengths on a NVIDIA A100 40GB. Both models are used with the torch.compile() module.

The model achieves better quality metrics than the original BERT while enjoying faster training and inference both in $O(N^2)$ and $O(N)$ regimes (Fig. 2).

To the best of our knowledge, we are the first to successfully train an NLP language model with no Softmax or any replacement/approximation for it in the attention layer. However, for vision tasks, such as object detection and instance segmentation, Zhuoran et al. (2021) propose two variations of attention, one without Softmax and the other with two softmaxes applied individually to Key and Query projections. However, they conduct experiments and report results only with second architecture. Recently, Koohpayegani & Pirsiavash (2024) instead scale Queries and Keys separately by their $l_1$ norm which allows them to successfully train a vision Transformer on ImageNet1K (Deng et al., 2009) and MS-COCO (Lin et al., 2014) datasets for different tasks with linear time complexity.

We opensource our code.

## 2 BACKGROUND

Here we give a brief exposition of essential elements of Transformer architecture and their variations.

Standard Transformer block consists of self-attention and feed-forward-network (FFN) sub-blocks (Vaswani et al., 2017). Let $\mathbf{X} \in \mathbb{R}^{N \times d}$, where $N$ is the sequence length and $d$ is an embedding dimension of one token. Define $\mathbf{Q} = \mathbf{X}\mathbf{W}_Q$ as queries, $\mathbf{K} = \mathbf{X}\mathbf{W}_K$ as keys, and $\mathbf{V} = \mathbf{X}\mathbf{W}_V$ as values, where $\mathbf{W}_Q, \mathbf{W}_K, \mathbf{W}_V \in \mathbb{R}^{d \times d_h}$ are learnable parameters. Then the *Scaled Dot-Product Attention* is formulated as:

$$\text{Attention}(\mathbf{X}) = \text{Attention}(\mathbf{Q}, \mathbf{K}, \mathbf{V}) = \text{Softmax}\left(\frac{\mathbf{Q}\mathbf{K}^\top}{\sqrt{d_h}} + \mathbf{M}\right)\mathbf{V}, \tag{1}$$

with Softmax applied row-wise and mask $\mathbf{M} \in \mathbb{R}^{N \times N}$ with values 0 or $-\infty$ which effectively disables some positions from calculation to account for causal sequence processing or to conceal 'PAD' token used for batch processing of sequences with different lengths.

Default implementation in some Transformer-based models (e.g. Devlin et al. (2019)) use biases in $\mathbf{Q}, \mathbf{K}$, and $\mathbf{V}$ projection layers.

Essentially, all transformer-based models use some form of Multi-Head Attention which has $H$ heads. Attention 1 is calculated for each head independently and the results are concatenated along the embedding dimension and projected back to full block's output dimension by a matrix $\mathbf{W}_O \in \mathbb{R}^{d \times d_{out}}$:

$$\text{MultiHeadAttn}(\mathbf{Q}, \mathbf{K}, \mathbf{V}) = \text{Concat}(\text{head}_1, \ldots, \text{head}_H)\mathbf{W}_O \tag{2}$$

Feed-Forward Network which follows self-attention is composed of two linear layers and an activation (usually ReLU or GeLU) in between. Intermediate inner dimension between the two layers is usually chosen to be 4x larger than input/ output dimension. Finally, a LayerNorm layer and a residual connection are applied around both blocks, their relative positions dictated by PreNorm or PostNorm architectural choice (Xiong et al., 2020). The formulation of the whole Transformer layer $l$ with PreNorm is:

$$\mathbf{X}'_l = \mathbf{X}_l + \text{Attention}(\text{LayerNorm}(\mathbf{X}_l))$$
$$\mathbf{X}_{l+1} = \mathbf{X}'_l + \text{FFN}(\text{LayerNorm}(\mathbf{X}'_l))$$

Thus, each full Transformer block has two LayerNorms and two residual connections.

Depending on the implementation, dropout (Srivastava et al., 2014) might also be used in various parts of the block, specifically after FFN and attention sub-blocks as in original Transformer, and in attention matrix before softmax as in BERT.

## 3  DESIGNING DENSEATTENTION

In this section, we describe the DenseAttention architecture and motivations that led to specific changes as compared to the Transformer. Then we outline two extensions aimed at adapting components widely and successfully used in contemporary models to the architecture: Cosine RelPE, and LocalAttention layers.

### 3.1  DENSEATTENTION

Since we aim to achieve as much computationally efficient and simple module as possible, we proceed with eliminating inefficient components of original self-attention and Transformer architectures.

The most straightforward idea which we exploit first is to abstain from using Dropout module anywhere in the model. Even though the module can be removed altogether at inference time, we also do it for the training as we believe it won't slow down the convergence with a large corpora dataset typical for LLM pre-training. Besides, as noted by Clark et al. (2019), dropout in attention probabilities might be the reason of redundancies among attention heads. Next, we remove the attention mask before Softmax. Note that if there are no biases in FFN, Query and Output linear layers and FFN activation is ReLU, then for a row vector $\mathbf{0}_d^\top = [0, 0, \ldots, 0]_{1 \times d}$

$$\text{Attention}(\mathbf{0}_d^\top \mathbf{W}_Q, \mathbf{K}, \mathbf{V}) = \mathbf{0}_d^\top \text{ and MultiHeadAttn}(\mathbf{0}_d^\top \mathbf{W}_Q, \mathbf{K}, \mathbf{V}) = \mathbf{0}_d^\top,$$

$$\text{FFN}(\mathbf{0}_d^\top) = \mathbf{0}_d^\top,$$

and

$$\text{LayerNorm}(\mathbf{0}_d^\top) = \mathbf{0}_d^\top,$$

i.e. zero vector stays intact when acted upon by all components of the Transformer module. So we refrain from using biases throughout the new block, fix representation of the "PAD" token at the output of embedding layer to $\mathbf{0}_d^\top$, and remove masking from the self-attention layer.

Subsequently, probably the most important modification that we impose on the old architecture is removal of row-wise Softmax activation from attention. We argue that the primary source of unparalleled modeling power of the original Transformer architecture which made it dominant architecture across multiple domains is the ability for all inputs to directly interact with each other in multiplicative way. This is the feature that all previous popular architectures like MLPs, CNNs and RNNs lack. We hypothesize that the role of softmax activation is ancillary to multiplicative interactions as it acts as a feature selection tool for the outputs of raw interactions matrix and normalizes them to be in $[0, 1]$ range and to add up to 1. We suggest these restrictions may be lifted without detrimental effect on performance.

However, removing Softmax proves to be a very challenging task exactly for this reason: without it attention outputs become unbounded which can lead for them to either diverge to $\infty$ or shrink to

0. We formalize this statement with the following proposition considering simplified version of the new mechanism where $\mathbf{W} = \mathbf{W}_Q \mathbf{W}_K^\top$ and $\mathbf{W}_V = \mathbf{I}$:

**Proposition 1.** *Let $\mathbf{X} \in \mathbb{R}^{N \times d}$ and $\mathbf{W} \in \mathbb{R}^{N \times d}$ be matrices composed of i.i.d. random variables, respectively $X_{ij}$ with $\mathbb{E}[X_{ij}] = 0$, $\mathrm{Var}(X_{ij}) = \sigma_X^2$, and $W_{km}$ with $\mathbb{E}[W_{km}] = 0$, $\mathrm{Var}(W_{km}) = \sigma_W^2$. Let $X_{ij}$ and $W_{km}$ also be independent for all $i, j, k, m$. Then each element of the matrix $\mathbf{Y} = \mathbf{X}\mathbf{W}\mathbf{X}^\top \mathbf{X} \in \mathbb{R}^{N \times d}$ has zero expectation and variance $\sigma_Y^2 \geq N d^2 \sigma_X^6 \sigma_W^2$.*

Essentially, it means that variance of an output grows at least as a cube of an input variance in the new architecture layer. And since $\sigma_Y^2$ along with tail probability $\mathbb{P}(|Y_{ij}| \geq t)$ are not bounded from above and depend on the form of an unknown distribution, we can't just fix $\sigma_X^2$ e.g. with the help of LayerNorm to ensure numerical stability.

Instead, we enforce $\max(|X_{ij}|) \leq a$ for some positive $a$ which is equivalent to setting fixed $L_\infty$ norm for the inputs. Consequently, even in worst case scenario where

$$X_{ij} = a \text{ for } \forall\, i, j \tag{3}$$

it holds for $\mathbf{Z} = \mathbf{X}\mathbf{X}^\top \mathbf{X} \in \mathbb{R}^{N \times d}$:

$$\max(|Z_{ij}|) \leq N d a^3, \tag{4}$$

i.e. $L_\infty$ norm of output values is bounded above. Furthermore, we make the following observation:

**Proposition 2.** *If elements $W_{km}$ of $\mathbf{W}$ are i.i.d normal variables with mean 0 and variance $\sigma_W^2$, independent with $\forall\, X_{ij}$, $\mathrm{Var}[(\mathbf{XW})_{pq}] \leq \sigma_W^2 a^2 d$*

It follows from **Prop. 2.** that $\sigma_W$ and $a$ can be chosen such that $\mathbb{P}[|(XW)_{pq}| \geq \epsilon] \leq \delta$ for some $\epsilon > 0, \delta > 0$ depending on $\sigma_W$ and $a$. Thus, we can assume that the matrix product $\mathbf{Y} = \mathbf{X}\mathbf{W}\mathbf{X}^\top \mathbf{X} \in \mathbb{R}^{N \times d}$ will not explode with right selection of priors.

Specifically, we set $a = \frac{1}{N^{\frac{1}{3}}}$, so that 4 becomes $\max(|Z_{ij}|) \leq d$. We choose not to downscale inputs by further degree, e.g. by $d\sqrt{n}$ because resulting small values may hurt modeling quality during training in low-precision formats (fp16 and bf16).

We fix each embedding vector $\mathbf{X}_i$ to have constant $l_\infty$ norm of 1 by applying our novel *MaxNormActivation* function:

$$\mathrm{MaxNormActivation}(\mathbf{X}_i) = \frac{\mathbf{X}_i}{\max_j(|\mathbf{X}|_{ij}) + \epsilon}$$

where $\epsilon$ is a very small number put to prevent division by 0. Note that similarly to *RMSNorm* (Zhang & Sennrich, 2019), *MaxNormActivation* doesn't center its inputs. However, it uses $l_\infty$ norm instead of $l_2$ and doesn't have *scale* and *bias* parameters as in Zhang & Sennrich (2019); Ba et al. (2016).

After *MaxNormActivation* we scale output by $\frac{1}{N^{\frac{1}{3}}}$. We acknowledge that both these calculations are memory bound but together they incur at most the same memory movement and compute cost as *LayerNorm*. In our ablation experiments any other activation or normalization function or absence thereof would lead to a prompt and unrecoverable numerical instability early on during training.

Consequently, it allows the removal of Softmax, which doesn't only lift a major computational and memory bottleneck which otherwise could be alleviated mainly with clever low-level algorithms as in Dao et al. (2022); Rabe & Staats (2021). Without Softmax and masking attention mechanism becomes a raw product of three matrices $\mathbf{Q}\mathbf{K}^\top \mathbf{V}$. Exploiting associative property of matrix multiplication, we can compute the product as

1. either $(\mathbf{Q}\mathbf{K}^\top)\mathbf{V}$ which yields $2N^2 d$ FMA operations,
2. or $\mathbf{Q}(\mathbf{K}^\top \mathbf{V})$ which yields $2N d^2$ FMA operations and is linear w.r.t $N$ both in time and memory complexity.

We can utilize both methods interchangeably depending on what's more favorable given particular values of $N$ and $d$. $O(N)$ complexity gives way to processing very large sequences in linear time with the same result as if done in traditional $O(N^2)$ paradigm as it calculates exactly the same all $N \times N$ pairwise interactions but just in another order.

Next, we consider reducing the number of heads in the multi-head attention as they are computationally inefficient. As extensive research efforts have shown (Bhojanapalli et al., 2020; Voita et al., 2019; Kovaleva et al., 2019; Michel et al., 2019), significant portion of heads in multi-head attention are redundant, output low-rank representations and can be pruned without decrease in quality in downstream tasks, at least in BERT-sized models. Specifically, Bhojanapalli et al. (2020) find that increasing number of heads past a certain threshold degrades performance in BERT. Motivated by this, we propose increasing $d_h$ from conventional value 128 up to 1024. In case of BERT example from C it leads to a single-head attention with arithm. int. 204.8 FLOPs/B which makes it computationally efficient even on NVIDIA A100. For LLMs with larger model dimension $d_h = 1024$ would still leave room for multiple heads. We also use $d_h = 256$ in experiments. And asymptotic arithm. int. in $O(N)$-regime is $\frac{d}{2}$ just like in an ordinary $d \times d$ dense layer.

We note that the matrix $\mathbf{W} = \mathbf{W}_Q \mathbf{W}_K^\top$ in the expression $\mathbf{Q}\mathbf{K}^\top = \mathbf{X}\mathbf{W}_Q \mathbf{W}_K^\top \mathbf{X}^\top$ is essentially low-rank as in standard attention $d_h \ll d$. But in our implementation this rank is much higher, in the extreme case being equal to $d$. It results in multiplication of two high or full rank matrices. That is a redundant operation from DL perspective because composition of linear maps is just another linear map which could be learned using half of the parameters. Thus, we decide to keep the $\mathbf{W}_Q$ and discard $\mathbf{W}_K$.

We also decide to remove LayerNorm and residual connection between attention and FFN sub-blocks as it improves computational efficiency of the architecture and appears not to hinder model performance. This leads to yet another simplification in the model design: $\mathbf{W}_V$ and $\mathbf{W}_O$ also become redundant by similar reasoning as in case of $\mathbf{W}_Q$ because there are no more non-linearities between attention outputs and FFN block.

Finally, the new attention mechanism in the case of a single head is formulated as:

$$\text{DenseAttention}\,(\mathbf{X}) = \mathbf{X}\mathbf{W}_Q \mathbf{X}^\top \mathbf{X} \in \mathbb{R}^{N \times d}$$

And in the case of multiple heads $H$ it slightly changes:

$$\text{DenseAttention}_h\,(\mathbf{X}) = \mathbf{X}\mathbf{W}_{Q_h} \mathbf{X}_h^\top \mathbf{X}_h \in \mathbb{R}^{N \times d_h}$$
$$\text{DenseAttention}\,(\mathbf{X}) = \text{Concat}_h[\text{DenseAttention}_h\,(\mathbf{X})]$$

We call our attention algorithm "DenseAttention" and the entire block as "DenseAttention Network" or DANet (spelled "dah-net") because it basically consists of dense matrix multiplications with little else. We notice that DenseAttention in multi-head setting resembles popular multi-query attention design from Shazeer (2019) as it also calculates different representations only for Queries.

To complete the DenseAttention Network, we apply *MaxNormActivation* and residual connection to outputs of FFN. Final architecture to the layer $l$ can be summarized as follows:

$$\mathbf{X}'_l = \text{DenseAttention}(\text{MaxNormActivation}(\mathbf{X}_l) \cdot N^{-\frac{1}{3}})$$
$$\mathbf{X}_{l+1} = \mathbf{X}_l + \text{MaxNormActivation}(\text{FFN}(\mathbf{X}'_l))$$

## 3.2 COSINE RELPE

Many modern Language Models use (Minaee et al., 2024) Rotary Positional Embeddings (RoPE) (Su et al., 2024) which evidently perform better than learned or sinusoidal positional embeddings and don't increase parameters count. The former two types of embeddings are applied once before the first layer and rely on skip-connections for propagating positional information to other layers in the stack. While it may be suitable for shallow networks, in deeper ones the signal gets decayed as more layers add their outputs to the residual branch. On the contrary, RoPE inject positional information into each of the Transformer layers by directly applying a transformation to the matrices $\mathbf{Q}$ and $\mathbf{K}$ which can be summarized as follows:

$$\mathbf{f}(\mathbf{x}_i, m) = \begin{bmatrix} \cos m\theta_i & -\sin m\theta_i \\ \sin m\theta_i & \cos m\theta_i \end{bmatrix} \begin{bmatrix} x_{i1} \\ x_{i2} \end{bmatrix},$$

where $\mathbf{x}_i = [x_{i1}\ x_{i2}]^T$ is a chunk $i$, $i \in \{0, \ldots, \frac{d}{2}\}$, of a vector $\mathbf{x}$ with $d$ dimensions which can be either a query $\mathbf{q}_m$ or key $\mathbf{k}_m$ with position $m$ out of $N$ in the sequence. Essentially, the transformation rotates the 2 two-dimensional vectors $\mathbf{q}'$ and $\mathbf{k}'$ with the intention to maximize their dot product

when they share the same position in sequence, and decay it to zero when the positions largely differ. However, direct calculation shows that it's not always true, as the result for some fixed $i$:

$$\mathbf{f}^\top(\mathbf{q}', m)\mathbf{f}(\mathbf{k}', n) = (q_1 k_1 + q_2 k_2)\cos(m-n)\theta + (q_2 k_1 - q_1 k_2)\sin(m-n)\theta \tag{5}$$

is only guaranteed to follow the pattern in case $\mathbf{q}'$ and $\mathbf{k}'$ are collinear. The total dot product of $\mathbf{q}$ and $\mathbf{k}$ is even less benign, for in each position $i$ of the model dimension, corresponding two-dimensional vector chunk has a possibly distinctive prior angle from the origin, and $\theta_i$ is also unique by construction:

$$\theta_i = 10000^{-2i/d}, \tag{6}$$

But Su et al. (2024) show that this parameterization leads to long-term decay in norm of attention scores with the increase of relative distance $m - n$.

Besides, RoPE are computationally inefficient as their calculation induces memory-expensive changes of tensor layout and several element-wise operations with low arithmetic intensity, separately for $\mathbf{Q}$ and $\mathbf{K}$. We notice that there exist two other transformations with more favorable efficiency properties which can be applied to scalars at individual positions $i \in \{0, \ldots, d\}$ of vectors $\mathbf{q}$ and $\mathbf{k}$ rather than paired numbers: $g_1(x_i, m) = x_i \cos m\theta_i$ and $g_2(x_i, m) = x_i(\cos m\theta_i - \sin m\theta_i)$. These produce similar expansions to 5:

$$g_1(q_i, m)g_1(k_i, n) = q_i k_i \cos m\theta_i \cos n\theta_i = q_i k_i[\cos(m-n)\theta_i - \sin m\theta_i \sin n\theta_i]$$
$$g_2(q_i, m)g_1(k_i, n) = q_i k_i[\cos(m-n)\theta_i - \sin(m+n)\theta_i]$$

We tested all three functions $\mathbf{f}$, $g_1$ and $g_2$ on LRA tasks with DenseAttention and found out that all of them impact the performance very similarly. However, when we set a constant $\theta$ for all positions in an embedding dimension, the quality dropped, adding evidence to the leading role of parameterization 6 in the RoPE potential.

We choose the simpler function $g_1$ as the new computationally efficient alternative to RoPE and name it *Cosine RelPE*. We use it extensively in conjunction with DenseAttention, however it can be readily applied to standard Transformer in place of RoPE.

We find that application of Cosine RelPE to $\mathbf{X}$ before DenseAttention layer, while affecting even matrix $\mathbf{X} = \mathbf{V}$ inside it, doesn't degrade the performance. Thus, we proceed with this architectural choice, which allows for one instead of two element-wise multiplications and can be further optimized by fusing with scaling factor $N^{-1/3}$.

## 3.3 LOCALATTENTION FOR DENSEATTENTION

In the years following invention of Transformer, many variations of *local attention*, also known as *sliding window attention*, patterns and implementations have been proposed (Zaheer et al., 2020; Beltagy et al., 2020; Child et al., 2019; Roy et al., 2021; Dao et al., 2022). Recently, some of the open-weights Large Language Models (Jiang et al., 2023; Team et al., 2024) started partially or fully adopting some forms of local attention with the primary goal of alleviating quadratic cost of full attention for large contexts with the trade-off of not being able to fully process the entire sequence at once.

We also develop a form of local attention pattern for discretionary use with DenseAttention on very long contexts, however, with the goal of improving modeling quality as opposed to increasing speed. The reason of this extension is outlined by Qin et al. (2022a): in linear Transformer family of models, attention scores of a query are distributed along the sequence length more uniformly as compared to Softmax attention, so the model is not fully able to focus at details in the vicinity of a query's token.

We adopt the approach to partition the whole sequence into equal non-overlapping chunks of *window size* $w$, similar to Dao et al. (2022); Qin et al. (2022a). We choose this design because of its simplicity and straight-forward implementation with minimal invocations of memory-intensive data layouts. However, this form of chunked attention leads to all of the tokens not being able to interact with up to a half of the tokens constituting their neighbourhood. To mitigate this issue, we extend our local attention framework beyond one layer and propose a 3-layer structure 3. It consists of LocalAttention, ShiftedLocalAttention, and global DenseAttention layers. The second,

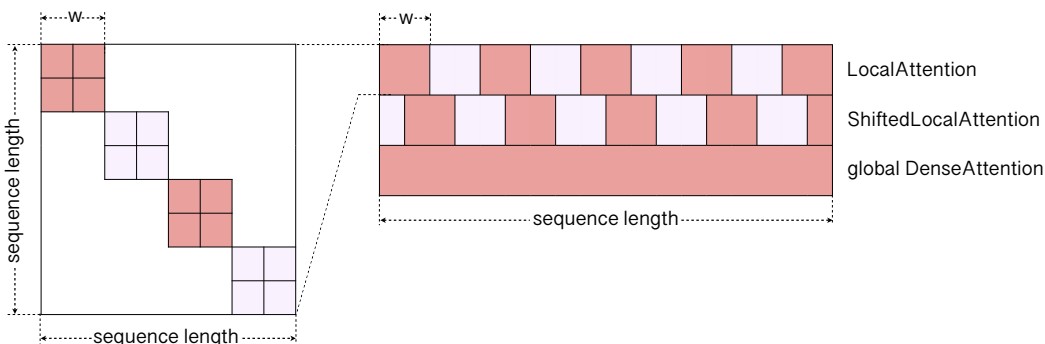

Figure 3: Local attention for DenseAttention scheme. Left: Chunked attention pattern of an individual local attention layer. Right: 3 layer structure of Local – LocalShifted – global attentions.

Table 1: Long Range Arena performance. Accuracy is the metrics for all benchmarks. Best results are in bold.

| Models | Listops | Text | Retrieval | Image | Pathfinder | PathX | Avg. |
|---|---|---|---|---|---|---|---|
| Transformers + Rotary | 47.90 | 79.08 | 82.31 | 75.04 | 76.64 | 84.72 | 74.28 |
| S4-v1 | **58.35** | 76.02 | 87.09 | **87.26** | 86.05 | 88.10 | **80.48** |
| DenseAttention | 50.50 | **81.19** | **87.51** | 72.55 | **87.40** | **88.82** | 77.99 |

ShiftedLocalAttention layer is shifted by $w/2$ relative to the first, which allows for all tokens to have symmetric neighbourhood after two consecutive layers. The full global attention of the last layer in the scheme combines fine-grained local results to capture all context of a sequence. The triples of layers then may be stacked together like ordinary Transformer layers to form a deep network.

We find local attention to be very effective in our experiments.

## 4 EXPERIMENTS

To prove the viability of DenseAttention architecture, we conduct two sets of experiments: 1) long range sequence modeling on Long Range Arena benchmark; 2) pretraining of BERT-like encoder architecture on sequences of different lengths. We train all of the models with fp16 precision, unless stated otherwise.

### 4.1 LONG RANGE ARENA

Long Range Arena is a challenging suite of 6 classification benchmarks dedicated to examining the abilities of efficient and long-context models on large sequence lengths spanning from 1k to 16k tokens. The tasks are diverse in nature and modalities: from synthetic and purely algorithmic, such as long version of ListOps benchmark (Nangia & Bowman, 2018), to character-level text classification on IMDB reviews (Maas et al., 2011). At the time of publication, the best model tested by Tay et al. (2021) achieved average of 55.01%, and all of the models failed to learn above the level of change on the most difficult task, Pathfinder-X (seq. len 16K) adopted from Linsley et al. (2018); Kim* et al. (2020).

Later, novel State-Space-Models-inspired architectures (Gu et al., 2022a;b; Ma et al., 2023; 2024) demonstrated by far superior performance considered to be out of reach for any Transformer-based model due specific inductive biases of the SSMs. But recently, Amos et al. (2024) showed that by using MLM-style pre-training the Transformer with RoPE is competitive with the SSMs. Interestingly, even without pre-training, but with RoPE, they reached a SOTA score on the benchmark among all Transformer-based architectures with a large margin.

Table 2: Ablations on the Retrieval task of LRA

| Model | Accuracy |
|---|---|
| DANet + Sinusoidal Embedding (bf16 format) | 82.69 |
| DANet + Cosine RelPE | 83.98 |
| DANet + Cosine RelPE + local attention (w=10) | **87.51** |

We take their scores as well as results from original S4 paper (Gu et al., 2022a) as two strong baselines and conduct extensive experiments on LRA dataset with DANet model to see if our architecture is capable of matching or surpassing them. We mostly follow specifications outlined in the original LRA paper including number of heads and model dimensions, adjusting sometimes number of parameters to the one used by Amos et al. (2024). We report the results in Table 1. DenseAttention Network establishes new SOTA score among the Transformer-based models and even outperforms the SSM in 4 out of 6 benchmarks.

Thus, we prove that DenseAttention architecture is competitive with standard attention even despite the simplifications, the absence of Softmax and the presence of non-smooth functions in the DANet architecture (MaxNorm and ReLU). We also show that Transformers can match the performance of SSMs in principle.

We use Cosine RelPE and Local-ShiftedLocal-Global attention scheme in all of LRA models. These extensions are useful for improving results which is exemplified in table 2. Local attention proves to be instrumental and, often, its window size is the most important hyperparameter to tune.

## 4.2 BERT PRETRAINING

We pre-train an encoder model with the approximately same number of parameters as in BERT-large (Devlin et al., 2019). We keep model dimension $d = 1024$ as in original work but increase number of layers from 24 to 32 to keep parity in number of parameters. We use the same MLM (Masked Language Modelling) + NSP (Next Sentence Prediction) combination of training objectives and pre-train on the same datasets, namely Wikipedia and BookCorpus (Zhu et al., 2015).

Table 3: Evaluations of MLM loss and accuracy for DenseAttention models w.r.t to BERT on C4 dataset. N is the maximum sequence length with which a model was trained or/and evaluated.

| Model | N=128 | | N=512 | | N=1024 | |
|---|---|---|---|---|---|---|
| | MLM Loss | Acc. | MLM Loss | Acc. | MLM Loss | Acc. |
| BERT-large | 2.67 | 0.561 | 2.42 | 0.59 | - | - |
| DenseAttention (1 head, N=128) | **2.13** | **0.577** | - | - | - | - |
| DenseAttention (1 head, N=512) | 2.19 | 0.572 | 1.92 | 0.603 | - | - |
| DenseAttention (1 head, N=1024) | 2.19 | 0.572 | **1.91** | **0.606** | 2.51 | 0.545 |
| DenseAttention (4 heads, N=128) | 2.19 | 0.568 | - | - | - | - |
| DenseAttention (4 heads, N=512) | 2.27 | 0.558 | 2.05 | 0.582 | - | - |
| DenseAttention (4 heads, N=1024) | 2.3 | 0.554 | 2.04 | 0.584 | **2.08** | **0.575** |

We pre-train two models: one with single head of size $d = 1024$ and the other with 4 heads of size $d = 256$. There are 4 training stages, each one resuming from the last checkpoint of the previous: first with approximately 850 million samples of sequence length 128, second with 150 mil. samples of seq. len 512, third with 80 mil. samples of seq. len 1024, and the last stage with 27 mil. samples of sequence length 16384 conducted exclusively with single head model. The single head model was trained in $O(N^2)$ regime with context sizes 128, 512 and, partially, 1024, and in $O(N)$ for the rest of the run with 1k and 16k contexts. The 4 heads model utilized the $O(N)$ regime for all sequence lengths.

Then we validate and compare the results with BERT-large, using Google's original pretrained checkpoint available from Hugging Face's Transformers library (Wolf et al., 2020). We evaluate

the models on out of domain texts of C4 dataset's subset "RealNewsLike" (Raffel et al., 2019) for all contexts lengths besides 16k because train/test splits for wiki + books dataset are almost surely different for our model and BERT training procedures. We use MLM loss which can be interpreted as logarithmic perplexity, and MLM accuracy as evaluation metrics. The results are presented in Table 3.

**Key highlights**. DenseAttention models uniformly outperform baseline in terms of MLM loss by a large margin. Perhaps, this may be contributed partially to dampening output logits (see appendix G) which lead to probabilities more calibrated to the ambiguity of natural language. Nevertheless, single-headed DenseAttention models also uniformly outperform standard BERT in terms of accuracy, although the difference is not so pronounced, as with log-perplexity.

The models with 4 heads are inferior in both metrics to single head ones which supports our hypothesis that larger head sizes lead to better quality. The only exception is the performance of the models trained with context length 1024 on sequences of the same size, where 4 heads DenseAttention model variant produces significantly better metrics. This might hint that it's easier for several heads to comprehend long sequences than for one. Note that the original BERT wasn't trained with sequence length 1024, so we couldn't compare it with our models in this setup.

Table 4: Throughput, sequences per second, of single head DenseAttention model in $O(N)$ and $O(N^2)$ regimes in comparison with BERT, and BERT with FlashAttention 2 across various sequence lengths. FLOPs ratio is total MatMul FLOPs of forward pass of DenseAttention BERT implementation in $O(N^2)$ regime divided by total MatMul FLOPs of forward pass of standard BERT. All experiments were conducted on a single NVIDIA A100 40Gb GPU.

| Seq. Len. | DenseAttention | | BERT | BERT with FlashAttn 2 | bs | FLOPs ratio |
|---|---|---|---|---|---|---|
| | $O(N)$ | $O(N^2)$ | | $O(N^2)$ | | $O(N^2)$ |
| 128 | 1403 | **1721** | 1450 | 1584 | 512 | 1.01 |
| 512 | 400.1 | **431.8** | 304.9 | 379.5 | 256 | 1.03 |
| 1024 | **208.9** | 208.8 | 117.9 | 181.6 | 128 | 1.05 |
| 2048 | **96.42** | 85 | - | 81.69 | 64 | 1.08 |
| 4096 | **48.09** | 33.38 | - | 33.93 | 32 | 1.13 |
| 8192 | **24.18** | 11.81 | - | 12.52 | 16 | 1.19 |
| 16384 | **13.47** | 4.1 | 0.943 | 4.12 | 8 | 1.24 |
| 32768 | **6.02** | 0.985 | - | 1.224 | 4 | 1.28 |
| 65536 | **3.03** | 0.378 | - | 0.338 | 2 | 1.30 |
| 131072 | **1.604** | - | - | 0.089 | 1 | 1.32 |

We also evaluate (Table 4) DenseAttention single head model speed, as measured by throughput, in comparison with standard BERT model and with highly-optimized, low-level FlashAttention-2 implementation which is the fastest conventional kernel for attention computation as of mid 2024 (Dao, 2024). All evaluations are performed using torch.compile() directive. As expected, DenseAttention model vastly outperforms even FlashAttention-2 algorithm with either quadratic or linear regime, depending on the sequence length. But, surprisingly, we also observed that with the increase of the sequence length the performance of the DenseAttetntion in the $O(N^2)$ regime is slightly worse or even similar to FlashAttention-2 despite being written in high-level language and having more FLOPs per iteration than a standard model with comparable size. It leads to conclusion that the DenseAttention indeed achieves very high computational intensity and FLOPs utilization in comparison with the alternatives.

Moreover, we observe that quality evaluation metrics stay the same for a fixed lengths validation context if the regime gets switched from $O(N)$ to $O(N^2)$ or vice versa regardless of the mode and sequence length with which a DenseAttention model has been trained. This invariance property holds even for the model trained on 16k context and applied to sequence length 128. Thus, we can train the models with DenseAttention on very large contexts in $O(N)$ time and then use it both short and long sequences with optimal speed and equal quality.

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

## A    CONCLUSION & FUTURE WORK

In this paper, we propose DenseAttention Network – a general architecture which simplifies the Transformer block and can serve as a drop-in replacement in every model architecture using it. We conduct experiments on the diverse modalities spanning from logic to language modeling and image classification and from short to extremely long sequence lengths using the LRA suite of benchmarks and MLM-style language model pre-training on text data. The results show that DenseAttention is capable of generalizing to many different tasks and context sizes and achieving favorable performance in comparison with standard Transformer and its augmented variants while being faster and more computationally efficient even with no specialized, low-level computation algorithms such as in Dao et al. (2022).

We acknowledge that there are other modalities and specialized architectures that would benefit from long-context efficiency improvements if the DenseAttention is ported or applied to them, such as ViT (Dosovitskiy et al., 2021) and SAM (Kirillov et al., 2023) for Computer Vision tasks, and LLAMA (Touvron et al., 2023a) for decoder-style language modeling. We hope to address them in future work. In particular, we look forward to adapting DenseAttention architecture to causal LLAMA-style LLMs and studying their scaling laws at billions parameter ranges.

## B    HARDWARE EFFICIENCY

All calculations performed by a hardware accelerator such as a NVIDIA GPU are either compute-bound or memory-bound (Williams et al., 2009). It depends on whether the operation in question spends the majority of time directly on computation or on data movements between High-Bandwidth Memory (HBM) and processing units. Customary unit of measurement for computational performance is TeraFLOPs (TFLOPs) per second and for memory it's bandwidth (throughput) in TB/s. Arithmetic intensity unifies both and is calculated as $\frac{\text{number of FLOPs}}{\text{number of bytes accessed}}$. It can be attributed both to hardware accelerator (usually referred to as *ops:byte ratio* in this case) and to a computational kernel, e.g. layer of neural network, and it's necessary but not sufficient for the kernel to maintain the arithmetic intensity higher than the accelerator in order to be computationally intensive (Docs, a). Otherwise, processing units stay idle part of the time waiting for the data to be brought from or written to HBM.

In latest generations of GPUs, FLOPs count rapidly grows but memory bandwidth progression falls behind, which results in latest generations of GPUs having much higher arithmetic intensity. Thus, it's increasingly hard for existing Deep Learning (DL) primitives to achieve hardware efficiency. Most operations besides matrix-matrix multiplications are inherently memory limited even on older GPUs. For example, the arithm. intensity of ReLU is 0.25 FLOPS/B, and for LayerNorm it's ¡ 10 FLOPS/B on NVIDIA V100 as stated in Docs (b). Moreover, GPUs feature fast Tensor Cores (312 TFLOPs for half-precision formats in NVIDIA A100) specialized for matrix multiplications, and general purpose cores with significantly lower throughput (19.5 TFLOPS in NVIDIA A100) which in turn process non-MatMul operations even slower as reported in He (2022).

So, from the view of computational efficiency, all activations, elementwise operations and reductions are detrimental to high ratios of hardware utilization.

## C    DISSECTING INEFFICIENCIES IN TRANSFORMER

Non-linearities, namely Softmax, LayerNorms, activation in FFN, dropouts, and skip-connections, which are present in Transformer architecture, indeed contribute majorly to its computational inefficiency, as documented in Ivanov et al. (2021); Pati et al. (2022); Portes et al. (2023). But other affine or linear transformations might also require further exploration. Consider two matrices $\mathbf{A} \in \mathbb{R}^{M \times N}$ and $\mathbf{B} \in \mathbb{R}^{N \times K}$ stored in half-precision floating point format which is common for DL apllications. Each element in the matrices has a size of 2 bytes, and each fused multiply-add (FMA) operation takes 2 FLOPs to compute (Docs, a). Then the arithmetic intensity of matrix multiplication in such setting is:

$$arithm.int._{MatMul} = \frac{M \cdot N \cdot K}{M \cdot N + N \cdot K + M \cdot K} \text{ FLOPs/B,} \tag{7}$$

as factors of 2 in the numerator and denominator both cancel out.

If there are no biases, then the two linear transformations in Transformer's FFN with model dimension $d$ and standard inner dimension $4d$ have arithm. int. of $\frac{4Nd}{5N+4d}$ which equals $\frac{4d}{5}$ as $N \to \infty$. $N$ dimension can accumulate both batch size $b$ and sequence length $s$ dimensions, and for BERT-large size model with $d = 1024, s = 512$, and $b = 128$ arithm. int. is approx. 809 FLOPs/B. For largest LLaMA 2 70B model with $d = 8192, s = 4096$, and $b = 1$ theoretical arithm. int. without using tensor parallelism (Narayanan et al., 2021) would be 2520 FLOPs/B. It's far greater than even NVIDIA H100 ops:byte ratio in both cases. Therefore, linear layers in the FFN are the most computationally efficient component of the Transformer and should be preserved in any hardware-aware architecture.

Similar argument may be applied to $K, Q, V$ projection layers in the self-attention, whose matrices can be concatenated together to yield $\frac{3d}{4}$ asymptotic arithm. intensity, and to the output projection by $W_O$ matrix in 2 ($\frac{d}{2}$ asymptotic arithm. int.). However, it follows from 7 that both products $\mathbf{S} = \mathbf{Q}\mathbf{K}^\top \in \mathbb{R}^{N \times N}$, and $\mathbf{O} = \mathbf{P}\mathbf{V} \in \mathbb{R}^{N \times d}$, where $\mathbf{P} = \text{Softmax}(\mathbf{S} / \sqrt{d_h} + \mathbf{M})$ have arithmetic intensity $\frac{N \cdot d_h}{N + 2d_h}$ with limit $d_h$ when $N \to \infty$. Also, batch and sequence dimensions cannot be fused for these operations because they are performed on *per sequence* level as opposed to *per embedding* level in FFN and KQV projections.

Large number of attention heads also contributes to inefficiency. Projection dimension of a head $i$ $\mathbf{Q}_i, \mathbf{K}_i$, and $\mathbf{V}_i$ is $\frac{d}{h}$ and typically equals 64 for smaller NLP language models like BERT, 256 for Google's PaLM (Chowdhery et al., 2022), and 128 for most others in the billions-parameters range, including LLAMA model family (Touvron et al., 2023a;b), Mistral (Jiang et al., 2023) and Mixtral 8x7B (Jiang et al., 2024), and GPT-3 (Brown et al., 2020).

Since the most common choice for $d_h$ is 128, the upper bound of arithm. int. of matrix multiplications inside attention mechanism is lower than even *ops:byte ratio* of an older V100 generation GPU. In the case of real-life configurations of BERT and LLaMA 2 from above the values are 32 and 120.5 FLOPs/B correspondingly. Thus, these operations are memory bound and inefficient.

So, from the computational perspective it is be beneficial to change number of heads in the attention to fewer or even single head with larger dimension $d_h$. Furthermore, it keeps the total number of flops constant because it equals $h \cdot N^2 \frac{d}{h} = N^2 d$ for all heads in total.

## D  SUB-QUADRATIC ALGORITHMS FOR SEQUENCE PROCESSING

Given entries $Q_i, K_j, V_j \in \mathbb{R}^{1 \times d}$ of matrices $\mathbf{Q}, \mathbf{K}$ and $\mathbf{V}$, standard softmax attention for input $i$ can be reformulated as

$$A_i = \frac{\sum_{j=1}^{N} \text{Sim}(Q_i, K_j) V_j}{\sum_{j=1}^{N} \text{Sim}(Q_i, K_j)} \in \mathbb{R}^{1 \times d},$$

where $\text{Sim}(Q_i, K_j) = \exp(Q_i K_j^\top)$. Conceptually, linear attention class of algorithms, described in Katharopoulos et al. (2020) and built upon in numerous subsequent works, approximates or replaces this similarity function with separable kernel $\text{Sim}(Q_i, K_j) = \mathcal{K}(Q_i, K_j) = \phi(Q_i)\phi(K_j^\top)$, where $\phi : \mathbb{R}^d \to \mathbb{R}_+^r$ maps query and key vectors to non-negative vectors with possibly different dimension $r$.

Hence, the attention mechanism becomes:

$$A_i = \frac{\sum_{j=1}^{N} \phi(Q_i)\phi(K_j^\top) V_j}{\sum_{j=1}^{N} \phi(Q_i)\phi(K_j^\top)} = \frac{\phi(Q_i) \sum_{j=1}^{N} \phi(K_j^\top) V_j}{\phi(Q_i) \sum_{j=1}^{N} \phi(K_j^\top)}, \tag{8}$$

which can be computed in linear time.

The function $\phi(\cdot)$ can take various forms, such as 1 + ELU (Katharopoulos et al., 2020), ReLU (Qin et al., 2022b), squared ReLU (Hua et al., 2022), Taylor (Duman Keles et al., 2023; Arora et al., 2024;

Zhang et al., 2024) or Random Feature (Choromanski et al., 2022; Peng et al., 2021) expansions, and even MLPs trained to mimic softmax attention (Zhang et al., 2024). They aim to approximate softmax without its explicit calculation when being applied jointly to queries and keys, or to retain its properties, most importantly, non-negativity of resulting dot products $\phi(Q_i)\phi(K_j^\top)$.

The latter property, together with reweighting attention scores (denominator in the formula 8) are defining for Linear Transformer algorithms. Absence of scaling by $\frac{1}{\phi(Q_i)\sum_{j=1}^{n}\phi(K_j^\top)}$ leads to numerical instabilities, and the scaling factor itself is not guaranteed to be bounded without non-negative $\phi(\cdot)$. However, both mappings $\phi(\cdot)$ (even relatively simple), and memory intensive non-MatMul operations for reweighting contribute to subpar speed and computational efficiency in comparison with ordinary and fast self-attention algorithms on all but large context sizes.

DenseAttention is substantially different from LinearTransformers. We forgo both transforming $\mathbf{Q}, \mathbf{K}$ by $\phi(\cdot)$ and reweighting in DenseAttention as we believe the main factor of success of Transformer is the ability of all $N \times N$ interactions between tokens. It results in an improved computational efficiency and simpler design which can be expressed entirely by matrix multiplications:

$$\mathbf{A} = \mathbf{Q}\mathbf{K}^\top\mathbf{V}$$

Another promising line of work focuses on applying deep State Space Models (SSMs) (Gu et al., 2022a; Gupta et al., 2022; Ma et al., 2023; Gu & Dao, 2024) and Linear RNNs (Beck et al., 2024; Orvieto et al., 2023; Peng et al., 2023) to long-range sequence and language modeling. Fundamentally, these architectures model interactions in sequence dimension by a linear recurrence:

$$x_t = \mathbf{A}x_{t-1} + \mathbf{B}u_t$$
$$y_t = \mathbf{C}x_t + \mathbf{D}u_t,$$

where $\mathbf{A}$ is some data-independent matrix which form and initialization are defining properties for a particular SSM/ RNN architecture. The linear recurrence is advantageous during inference as it runs in $O(N)$ time. For training, it also can be unrolled into a convolutional kernel

$$\mathbf{K} = \begin{bmatrix} \mathbf{CB}, & \mathbf{CAB}, & \dots, & \mathbf{CA}^{L-1}\mathbf{B} \end{bmatrix}$$

to compute

$$y = \mathbf{K} * u.$$

via Fast Fourier Transform (FFT) in $O(\log n)$ time.

While being sub-quadratic, these algorithms are still slower than linear time as in DenseAttention. However, recently Gu & Dao (2024) introduced data-dependent gating in $\mathbf{A}$ and low-level, hardware efficient CUDA implementation for parallel-scan operation which allows for fast linear-time processing both during training and inference.

# E  THE LRA BENCHMARK

## E.1  DISCUSSION OF THE LRA TASKS

The Long Range Arena is, in fact, not a single benchmark but a suite of 6 challenging and diverse tasks designed to test modeling capabilities across different domains. Below is a brief description of each task.

**ListOps**(Nangia & Bowman, 2018). This is a purely logical synthetic task which is dedicated to modeling evaluation results of long hierarchically structured sequences. Each sequence has length up to 2000 symbols and consists of whole numbers from 0 to 9, mathematical operators, such as MAX, MIN, MEDIAN and SUM_MOD, and parentheses.

**Text Classification (IMDB)** (Maas et al., 2011). This task tests Natural Language Understanding (NLU) abilities of models by letting them classify the sentiment of movie reviews in the IMDB dataset. To make the task more challenging, the texts of the reviews are split into tokens not on a word level, but on a character (or byte) level. This leads to much longer sequences of 4K max length.

**Document Retrieval (AAN)** (Radev, 2013). This task tests the abilities of producing encoded representations of the textual information and further matching/ retrieving them. Namely, given a pair of the documents from ACL Anthology Network (AAN; Radev et al., 2013) dataset, a model should independently process them and, based on their final embeddings, classify if the two documents have a citation link. As in the IMDB tasks, individual input texts are tokenized on a character (byte) level with max sequence length 4K.

**Image Classification (CIFAR-10)** (Krizhevsky & Hinton, 2009). This is an image classification task with 10 classes on a classical CIFAR-10 benchmark with one specific condition: images should be ingested into models as 1-d sequences, thus setting the input length to 1024 tokens (pixels) and making the task more challenging.

**Pathfinder** (Kim* et al., 2020) . This is a binary classification task of 32x32 pixels grayscale images with corresponding sequence length 1024 tokens, which, formally, makes it similar to CIFAR-10 task. However, it's different on a conceptual level, as the task measures a model's ability to discern spatial dependencies. Given a multitude of intertwined, dashed line paths, a model should correctly determine if two rounded dots are connected by a dashed line.

**Pathfinder-X (Pathfinder-128)**. It's a version of Pathfinder task with 16K (128x128) pixels images which makes it significantly more challenging. At the time of publication of the original LRA paper Tay et al. (2021), none of the tested models managed to achieve a score above chance on this benchmark.

Therefore, the Long Range Arena arguably represents a wide range of tasks, spanning from logic and reasoning to language modeling and image classification. To perform well on all of the 6 benchmarks, a model's architecture should be powerful and versatile enough to generalize to different modalities.

## E.2  EXTENDED COMPARISONS WITH TRANSFORMER-BASED MODELS

Full comparisons with an exhaustive list of Transformer-based models which, to the best of our knowledge, have been tested on the LRA, including the most recent ones, show that DenseAttention outperforms all of them.

Table 5: Long Range Arena performance. Accuracy is the metrics for all benchmarks. Best results are in bold and second best are underscored. To ensure consistent comparisons, the averages for the models which report the result on Path-X task are computed without it.

| Model | Listops | Text | Retrieval | Image | Pathfinder | PathX | Avg. |
|---|---|---|---|---|---|---|---|
| Transformer (Tay et al., 2021; Dao et al., 2022) | 36.37 | 64.27 | 57.46 | 42.44 | 71.40 | 61.40 | 54.39 |
| Local Attention (Tay et al., 2021) | 15.82 | 52.98 | 53.39 | 41.46 | 66.63 | - | 46.06 |
| Sparse Trans. (Tay et al., 2021) | 17.07 | 63.58 | 59.59 | 44.24 | 71.71 | - | 51.24 |
| Longformer (Tay et al., 2021) | 35.63 | 62.85 | 56.89 | 42.22 | 69.71 | - | 53.46 |
| Linformer (Tay et al., 2021) | 35.70 | 53.94 | 52.27 | 38.56 | 76.34 | - | 51.36 |
| Reformer (Tay et al., 2021) | 37.27 | 56.10 | 53.40 | 38.07 | 68.50 | - | 50.67 |
| Sinkhorn Trans. (Tay et al., 2021) | 33.67 | 61.20 | 53.83 | 41.23 | 67.45 | - | 51.29 |
| Synthesizer (Tay et al., 2021) | 36.99 | 61.68 | 54.67 | 41.61 | 69.45 | - | 52.88 |
| BigBird (Tay et al., 2021) | 36.05 | 64.02 | 59.29 | 40.83 | 74.87 | - | 55.01 |
| Linear Trans. (Tay et al., 2021) | 16.13 | 65.90 | 53.09 | 42.34 | 75.30 | - | 50.55 |
| Performer (Tay et al., 2021) | 18.01 | 65.40 | 53.82 | 42.77 | 77.05 | - | 51.41 |
| RFA (Peng et al., 2021) | 36.80 | 66.00 | 56.10 | - | - | - | - |
| Luna-256 (Ma et al., 2021) | 37.98 | 65.78 | 79.56 | 47.86 | 78.55 | - | 61.95 |
| Nyströmformer (Xiong et al., 2021) | 37.15 | 65.52 | 79.56 | 41.58 | 70.94 | - | 58.95 |
| Kernelized Attention (Chen et al., 2021) | 38.78 | 60.22 | 81.77 | 41.29 | 70.73 | - | 58.56 |
| Informer (Chen et al., 2021) | 32.53 | 62.64 | 77.57 | 38.10 | 57.83 | - | 53.73 |
| Skyformer (Chen et al., 2021) | 38.69 | 64.70 | 82.06 | 40.77 | 70.73 | - | 59.39 |
| cosFormer (Qin et al., 2022b) | 37.90 | 63.41 | 61.36 | 43.17 | 70.33 | - | 55.23 |
| FNet (Lee-Thorp et al., 2022) | 35.33 | 65.11 | 59.61 | 38.67 | 77.80 | - | 55.30 |
| FLASH-quad (Qin et al., 2022a) | 42.20 | 64.10 | 83.00 | 48.30 | 63.28 | - | 60.18 |
| FLASH (Qin et al., 2022a) | 38.70 | 64.10 | 86.10 | 47.40 | 70.25 | - | 61.31 |
| TransNormer T1 (Qin et al., 2022a) | 41.03 | 66.90 | 83.11 | 51.60 | 75.92 | - | 63.71 |
| TransNormer T2 (Qin et al., 2022a) | 41.60 | 72.20 | 83.82 | 49.60 | 76.80 | - | 64.80 |
| KDEformer (Zandieh et al., 2023) | 36.64 | 62.00 | 73.52 | 45.45 | 68.13 | - | 57.15 |
| Hedgehog (Zhang et al., 2024) | 37.15 | 64.60 | 82.24 | 40.15 | 74.16 | - | 59.66 |
| Transformers + Rotary (Amos et al., 2024) | 47.90 | 79.08 | 82.31 | **75.04** | 76.64 | 84.72 | 72.89 |
| DenseAttention (ours) | **50.50** | **81.19** | **87.51** | 72.55 | **87.40** | **88.82** | **75.83** |

## F PROOFS

**Proof of Proposition 1:**

$$Y_{ij} = \sum_{n=1}^{N} \sum_{m=1}^{d} \sum_{k=1}^{d} X_{ik} W_{km} X_{mn}^{\top} X_{nj}$$

Denote $S(i; k; m; n; j) = X_{ik} W_{km} X_{mn}^{\top} X_{nj}$. Since $\mathbb{E}[W_{km}] = 0$ and $W_{km}$ is independent from $X$, $\mathbb{E}[S(i; k; m; n; j)] = 0$ and $\mathbb{E}[Y_{ij}] = \sum_{k,m,n} \mathbb{E}[S(i; k; m; n; j)] = 0$. Hence, $\text{Var}[S(i; k; m; n; j)] = \mathbb{E}[X_{ik}^2 W_{km}^2 (X_{mn}^{\top})^2 X_{nj}^2] - 0$.

As some of the indices $i, k, m, n, j$ can be the same number, there are three possible options for $\text{Var}[S(i; k; m; n; j)]$:

1. $\mathbb{E}[x_1^2 x_2^2 x_3^2] \mathbb{E}[w^2] = \sigma_X^6 \sigma_W^2$ by independence of all $x$ and $w$.
2. $\mathbb{E}[x_1^4 x_2^2] \mathbb{E}[w^2] = \mathbb{E}[x_1^4] \mathbb{E}[x_2^2] \sigma_W^2 \geq \sigma_X^6 \sigma_W^2$, because by Jensen's inequality $\mathbb{E}[g(x^2)] \geq g(\mathbb{E}[x^2])$ and we let $g(f) = f^2$.
3. $\mathbb{E}[x^6] \mathbb{E}[w^2] \geq \sigma_X^6 \sigma_W^2$ by similar reasoning ($g(f) = f^3$ is convex on $(0, \infty)$).

Finally, $Cov(S_p, S_q) = 0$ if the set of indices $p$ is not identically equal to set $q$ because even one distinct index between $p$ and $q$ leads to independent factors inside the covariance operator. Therefore, $\text{Var}[Y_{ij}] \geq N d^2 \sigma_X^6 \sigma_W^2$. $\qquad\square$

**Proof of Proposition 2:** If we let $\mathbf{X}_{ij} = a$ be a degenerate R.V. as in worst case, 3, then $\text{Var}[(\mathbf{XW})_{pq}] = \sigma_W^2 a^2 d$ by C.L.T and properties of variance. In all other cases, from $X_{ij} \in [-a, a]$ follows that $\sigma_{X_{ij}}^2 \leq a^2$ by Popoviciu's inequality (Popoviciu, 1935). Then $\text{Var}[X_{pj} W_{jq}] = \sigma_{X_{pj}}^2 \sigma_{W_{jq}}^2 \leq a^2 \sigma_W^2$, and $\text{Var}[(\mathbf{XW})_{pq}] = \sum_{j=1}^{d} \text{Var}[X_{pj} W_{jq}] \leq \sigma_W^2 a^2 d$ even if some $X_{pj}$ is dependent with some $X_{pj'}$, because $Cov[\sigma_{X_{pj}}^2 \sigma_{W_{jq}}^2; \sigma_{X_{pj'}}^2 \sigma_{W_{j'q}}^2] = 0$ for $j \neq j'$. $\qquad\square$

## G  DETAILS OF THE BERT TRAINING PROCEDURE AND RESULTS ON 16K CONTEXT.

To ensure numerical stability, we scale weight matrices of FFN layers to have a constant $l_\infty$ norm after each optimizer step during pre-training. After pretraining, we merge each weight with its final scaling factor so there is no additional overhead at the inference time. The choice of the norm type is motivated largely by the bounds it provides for the layer outputs as in the case with the DenseAttention layer. The scaling factor of a layer is a standalone non-trainable scalar decoupled from its corresponding weight tensor at the train time. This means that the weight itself doesn't get re-scaled constantly which would otherwise induce tug-of-war dynamics with the direction of gradient. This way, the weight also has natural proportions compared to ADAM optimizer's ((Kingma & Ba, 2015)) weight update as it would in the absence of scaling. By employing this technique, we eliminate the need for weight decay and warmup. We also use constant learning rate $2 \times 10^{-4}$ in all training runs.

We observed that scaling the Queries weight in the DenseAttention hinders loss convergence speed to a certain degree so we proceeded with scaling just FFN layers.

The models which continued training with seq. len. 1k slightly outperform their counterparts which stopped after seq. len. 512. on sequences of this same size 512 which indicates that training on longer contexts is indeed beneficial for modeling quality. However, performance degrades when models trained on $N = 512$ or $N = 1024$ get tested on seq. len. 128 which is a consequence of the models' specialization on the longer sequences.

This property gets even more noticeable with the single head model trained on 16k context (Table 6). The pre-training was performed on the dataset which contains 26% of sequences with max size of 16k tokens, and 45% with size $\geq$ 1024. Therefore, the model is adapted to long-context and performs much better in terms of evaluation metrics on the datasets and context lengths with greater maximum size. We argue that quality metrics of the model trained on 16k context size, while inferior to the metrics of the smaller-context checkpoints on their respective lengths, is actually quite impressive, as it correctly finds the right token out of 30.5 thousand vocabulary options 45% of the time for approximately 2000 masked tokens in a single sequence of size 16k. And with the decrease of the context length to 2048 tokens, the model quality becomes almost equal to the smaller-context models evaluated with their native sequence sizes.

Table 6: Quality metrics for single-head DenseAttention model trained on the context of up to 16k tokens. Books dataset contains > 98% of sequences with length > 1024, and for each tested max. seq. len. it's guaranteed to contain at least. 80% of sequences with such length. C4 dataset for max seq. lengths 1024 and 2048 has approx. 9.5% sequences with context size $\geq$ 1024.

| max seq. len. | Books | | C4 | |
|---|---|---|---|---|
| | MLM loss | acc. | MLM loss | acc. |
| 16384 | 2.76 | 0.451 | - | - |
| 8192 | 2.64 | 0.482 | - | - |
| 4096 | 2.45 | 0.511 | - | - |
| 2048 | 2.21 | 0.549 | 2.4 | 0.545 |
| 1024 | - | - | 2.59 | 0.506 |
| 512 | - | - | 2.55 | 0.513 |

We code the model in plain PyTorch (Paszke et al., 2019) and train it in distributed mode using Deep-Speed (Rasley et al., 2020) in fp16 precision, using the framework's native implementation which is similar to NVIDIA's AMP (Micikevicius et al., 2018). We found out during ablation experiments that training in bf16 format converges significantly slower, likely because it has less precision bits than fp16. bf16 also has a disadvantage that it doesn't work on older GPUs such as NVIDIA V100.

# H ADDITIONAL EXPERIMENTS

## H.1 PATHFINDER-256

Pathfinder-256 is an extremely challenging version of the Pathfinder task with sequence length 65k which is on par with input context size of recent generations of proprietary Large Language Models.

Table 7: Accuracy on Pathfinder-256 task

| Algorithm | Accuracy on the validation set, % |
|---|---|
| FlashAttention (Dao et al., 2022) | 63.1 |
| S4 (Amos et al., 2024) | 67.8 |
| DenseAttention | 72.6 |
| DenseAttention after additional 550 epochs | 77.1 |

DenseAttention model outperforms (Table 7) existing results from the literature of standard Transformer augmented with FlashAttention (Dao et al., 2022) and S4-v2 model (Gu et al., 2022b) as reported in Amos et al. (2024). The result holds both when the training procedure is carried out for 200 training epochs as in Dao et al. (2022) and then it's prolonged for 550 additional epochs.

This experiment lets us make several observations:

- DenseAttention Network architecture performs well even on very long input sequences which is promising given current trend of increasing context size in modern Large Language and Multimodal Models;

- DenseAttention shows favorable scaling properties with respect to the amount of training iterations, even with the fixed dataset size. The validation accuracy for the task kept improving throughout the whole training and would likely have continued if the experiment had not been stopped;

- Truly linear scaling in sequence length is crucial for improvements in quality for large contexts. It took approximately 3 days on 4 H100 GPUs to train our model for 750 epochs in linear mode, while the projected runtime of quadratic FlashAttention-2 (Dao, 2024) and log-linear (S4) algorithms in the same setting would be at best 3 and 0.5 months, respectively, which renders them impractical for prolonged training.

## H.2 ABLATION STUDY ON RELPE

Regular Rotary Positional Embeddings (RoPE) (Su et al., 2024) are known to enhance modeling performance and generalization in Transformer models and are widely used (Biderman et al., 2023; Black et al., 2022; Chowdhery et al., 2022; Dubey et al., 2024). In fact, just by incorporating it into a standard Transformer model, Amos et al. (2024) managed to beat all efficient and long-context modifications of Transformer on the Long Range Arena benchmark.

Table 8: Ablation on RelPE. Comparison of training and inference speeds (in sequences per seconds) on the LRA's Pathfinder task.

| Model variant | Training Speed, (speed-up) | Inference Speed (speed-up) |
|---|---|---|
| Rotary Embeddings | 7025 (1.00x) | 16908 (1.00x) |
| Cosine Embeddings q,k | 10276 (1.46x) | 28467 (1.68x) |
| Cosine Embeddings | 10438 (1.49x) | 29630 (1.75x) |

However, regular RoPE are not computationally efficient. Our primary motivation behind designing Cosine RelPE is speed and efficiency gains, as we aimed to make DenseAttention as efficient as possible. As we demonstrated in the paper, expanded expressions for RoPE and Cosine RelPE are similar while the latter form of embeddings involves much less memory-intensive computations. Empirically, we found that the difference in modeling quality between the two types is negligible.

We present the results of the ablation study on speed in the table 8. Cosine RelPE are significantly faster in both scenarios. "q, k" in the second row denotes that Cosine RelPE were applied separately to Q and K matrices like in regular RoPE.

## H.3 SCALING EFFECT STUDY

Table 9: Scaling study on DenseAttention-BERT architecture

| Model | Parameters | Configuration | MLM loss | MLM accuracy |
|---|---|---|---|---|
| DANet-BERT-small | 31M | L=6, D=512 | 2.74 | 49.5 |
| DANet-BERT-base | 110M | L=16, D=768 | 2.02 | 60.0 |
| DANet-BERT-large | 336M | L=32, D=1024 | 1.70 | 64.9 |

The table H.3 depicts three single-head DenseAttention Network models of different sizes pre-trained on Wiki+BookCorpus dataset with MLM objective for 100B tokens. MLM loss and accuracy are reported for out-of-sample data from C4 dataset (Raffel et al., 2019). L and D parameters denote number of layers and hidden dimension of FFN input, respectively.

