# OpenReview forum: "DenseAttention: No-Compromise Exact All $N \times N$  Interactions Algorithm with $O(N)$ Space and Time Complexity"
_ICLR.cc/2025/Conference — Submitted to ICLR 2025_

### Official Review · Reviewer_rVwA · 2024-10-20

**Soundness:** 2
**Presentation:** 2
**Contribution:** 2
**Rating:** 5
**Confidence:** 4

**Summary:**

The authors propose a new architecture which they call DenseAttention Network, which is a variation of the standard transformer architecture which is specifically tuned to perform well on long sequences. The changes include:
1) Take the softmax away in the attention block, writing $QK^{\top}V$ instead of $\mathrm{softmax}(QK^{\top})V$, and using associativity to compute the matrix product in linear time. (They name this as DenseAttention mechanism/block.)
2) Use a MaxNormActivation block instead of LayerNorm, which scales each token feature by its maximum absolute value.
3) Use a novel positional embedding called Cosine RelPE, which is claimed to perform similarly but more efficiently computable than RoPE.
4) For very long contexts, use a hand-rolled local attention implementation suited for DenseAttention.
They show that this architecture has general improvements over a basic transformer + RoPE implementation in Long Range Arena, both in performance (across a few tasks) and efficiency (more broadly, as the usual attention mechanism does not have linear-in-$N$ time complexity). It also shows improvements against S4-V1 in Long Range Arena, and BERT in terms of Masked Language Modeling.

**Strengths:**

- The MaxNormActivation block is potentially useful as a method to stabilize LLM training.
- The empirical results in tables 1-3 show that DenseAttention has some promise empirically on long-context sequence modeling tasks as compared to the standard transformer, S4-V1, and BERT.
- The efficiency results in table 4 show that it's possible to get out-of-the-box performance increases at long context compared to the usual BERT model. Specifically, all changes seem to be architectural, and no specialized kernels are needed to get better performance, as a result of using `torch.compile` and potentially fusing linear operations together.

**Weaknesses:**

- The main mechanism behind DenseAttention, i.e., removing the softmax and using associativity to compute the product in linear-in-$N$ time, has been studied before; see for example [1]. It is acknowledged that the paper cites [1], but the paper suggests that the mechanism in [1] has poor efficiency; however DenseAttention is strictly a special case of the mechanism in [1] (using the notation of [1], set $\phi$ as the identity mapping). So this claim does not make sense, and the novelty of DenseAttention seems limited.

- The reasoning behind using MaxNormActivation seems lacking. In particular, since all norms are equivalent (i.e., bounded by each other up to multiplicative constants, possibly dependent on dimension) in finite-dimensional vector space, the boundedness of the maximum norm is equivalent to the boundedness of the $\ell^{2}$ norm. So if the argument in the paper goes through, it should mean that $\ell^{2}$ normalization should also work (then why not LayerNorm? But LayerNorm doesn't work, as reported in the paper, so something else is going on.). Although the MaxNormActivation is an interesting and potentially useful contribution, it may not work for the reason explained in the paper. Also there's a potential typo in the equation defining MaxNormActivation: it should be $\frac{X_{i}}{\max_{j}|X_{ij}| + \epsilon}$ on the RHS (note the absolute value).

- Not much motivation is given for the two other modifications, e.g., CosineRelPE and the local attention proposal - they seem to have a flavor of "we tried it and it works," potentially with some ablation, and without context of why such an approach may or may not make sense or generalize to other architectures.

- The results in Long Range Arena are promising insofar as they match up against a standard transformer and an SSM, but this may not be a fair comparison. Given that the authors start with a regular transformer and apply modifications to show improvement on long-context, they could also compare against more recent models specialized for long context. For example, the authors omit comparison with S5, whose numbers are publicly available on [PapersWithCode](https://paperswithcode.com/dataset/lra), as well as S4 V2 and a long list of other models benchmarked on Long Range Arena but not necessarily added there.

- The result on efficiency compared to BERT also may seem to not be a fair comparison. BERT is trained with an encoder-only architecture, while DenseAttention Network is trained with a decoder-only architecture. A fairer comparison would pit DenseAttention Network against a regular decoder-only transformer (as well as BERT if desired, along with, say, an SSM), under the same experimental setting, and allow readers to observe trends in the different approaches as different scaling parameters vary.


[1] Katharopoulos, Angelos, et al. "Transformers are RNNs: Fast autoregressive transformers with linear attention." International conference on machine learning. PMLR, 2020.

**Questions:**

- What is the specific motivation of designing CosineRelPE?

- Is there anything that suggests that any new block (DenseAttention, CosineRelPE, MaxNormActivation) can generalize to other architectures and improve either performance or efficiency (while not degrading the other)?

---

> ### Author Response · Authors · 2024-11-27
> **Author response. Part 1**
>
> We thank the reviewer rVwA for valuable review and thought-provoking comments. We are delighted by your appreciation of both modeling quality on long contexts, as well as performance and efficiency gains of DenseAttention, which were the main reasons for much inner working behind the DANet architecture. Please let us address your concerns.
>
> **Weakness 1**
>
> > The main mechanism behind DenseAttention, i.e., removing the softmax and using associativity to compute the product in linear-in-$N$ time, has been studied before; see for example [1]. It is acknowledged that the paper cites [1], but the paper suggests that the mechanism in [1] has poor efficiency; however DenseAttention is strictly a special case of the mechanism in [1] (using the notation of [1], set  $\phi$ as the identity mapping). So this claim does not make sense, and the novelty of DenseAttention seems limited.
>
> ---
>
> Motivated by your and other reviewers’ comments, we wrote a detailed exposition of the Linear Transformer class of algorithms and their fundamental differences with DenseAttention, which we will include in the revision of the paper. We share it in the General Response and gently ask you to read it. We would also like to address specific points in your comment.
>
> As we discuss there, DenseAttention Network’s architecture is quite different from the Linear Transformer. The mandatory building blocks of their architecture (section 3.2 in [1]) and numerous derivative works are non-negative mapping $\phi(\cdot)$  and reweighting scheme implemented as denominator in $\frac{\phi(Q_i) \sum_{j=1}^N  \phi(K_j^\top)V_j}{\phi(Q_i) \sum_{j=1}^N  \phi(K_j^\top)}$. Both elements are memory-intensive operations and contribute to computational inefficiency of Linear Transformers similar to regular softmax attention.
>
> We don’t utilize any of these blocks and empirically show that modeling quality is, in fact, better.
>
> Framing the absence of any transformations applied to $\mathbf{Q}$ and $\mathbf{K}$ as an identity mapping $\phi(x)=x$ would be conceptually wrong because transformed values are required to be non-negative in the Linear Transformer class of algorithms [1-2]. And the absence of reweighting the attention scores by their row-wise sums further sets DenseAttention apart from Linear Transformers.
>
> In fact, getting rid of attention reweighting was the hardest part in designing the new algorithm because without it attention outputs tend to diverge to infinity very quickly in real-world scenarios even with moderately deep networks and moderately large head sizes. It became the primary reason for using MaxNormActivation.
>
> ---
>
> **Weakness 2**
> > The reasoning behind using MaxNormActivation seems lacking. In particular, since all norms are equivalent (i.e., bounded by each other up to multiplicative constants, possibly dependent on dimension) in finite-dimensional vector space, the boundedness of the maximum norm is equivalent to the boundedness of the $\ell^{2}$ norm. So if the argument in the paper goes through, it should mean that $\ell^{2}$ normalization should also work (then why not LayerNorm? But LayerNorm doesn't work, as reported in the paper, so something else is going on.). Although the MaxNormActivation is an interesting and potentially useful contribution, it may not work for the reason explained in the paper.
>
> We thank you for this very interesting comment and are happy give detailed explanations.
>
> Without reweighting, the attention mechanism becomes just a composition of matrix multiplications: $\mathbf{A}=\mathbf{X} \mathbf{W} \mathbf{X}^\top \mathbf{X}$. Outputs in this expression grow cubically w.r.t inputs: for any element $X_{i, j}$ of input matrix $\mathbf{X}$, there exists an output element $A_{m, n}$, which has $X_ {i, j}^3$ as a summand. It can lead to exploding or vanishing output values, depending on the distribution of the inputs, especially when computed in half-precision formats (fp16’s max value is just 65504). Hence, it becomes imperative to control the magnitude of inputs.
>
> Naturally, the most suitable norm for directly controlling the maximal magnitude is $\ell_\infty$. Although it’s true that $\lVert x \rVert_\infty \leq \lVert x \rVert_2$ for a finite-dimensional vector $x$, scaling $x$ by $\frac{1}{\lVert x \rVert_\infty}$ is preferable because it guarantees that the absolute maximum value of its elements is exactly 1. It prevents outputs from both exploding to $\infty$ or $NaN$ values or shrinking to 0.
>
> Furthermore, calculation of $\lVert x \rVert_2$ in low precision formats can lead to either a numerical instability (as in the case of fp16) or to a loss of numerical precision (both in fp16 and bf16) for high-dimensional $x$.
>
> Finally, finding maximum absolute value in a vector is relatively cheaper computationally than squaring, adding all the elements and then taking a root.

---

> ### Author Response · Authors · 2024-11-27
> **Author response. Part 2**
>
> > Also there's a potential typo in the equation defining MaxNormActivation: it should be $\frac{X_{i}}{\max_{j}|X_{ij}| + \epsilon}$ on the RHS (note the absolute value).
>
> Thank you for noting this typo. We will fix it in the next revision.
>
> ---
>
> **Weakness 3**
>
> > Not much motivation is given for the two other modifications, e.g., CosineRelPE and the local attention proposal - they seem to have a flavor of "we tried it and it works," potentially with some ablation, and without context of why such an approach may or may not make sense or generalize to other architectures.
>
> ---
>
> We thank you for pointing out the perceived lack of motivation behind designing CosineRelPE and the local attention.
>
> Regular Rotary Positional Embeddings (RoPE) [3] are known to enhance modeling performance and generalization in Transformer models and are widely used [4-7]. In fact, just by incorporating it into a standard Transformer model, Amos et al. [8] managed to beat all efficient and long-context modifications of Transformer on the Long Range Arena benchmark.
>
> However, regular RoPE are not computationally efficient as we explain in section 3.2 of the paper. Our primary motivation behind designing Cosine RelPE is speed and efficiency gains, as we aimed to make DenseAttention as efficient as possible. As we demonstrated in the paper, expanded expressions for RoPE and Cosine RelPE are similar while the latter form of embeddings involves much less memory-intensive computations. Empirically, we found that the difference in modeling quality between the two types is negligible.
>
> Motivated by your comments, we conducted an ablation study on speed. We present the results below.
>
>
> | Model variant          | Training Speed (speed-up) | Inference Speed (speed-up) |
> |------------------------|---------------------------|----------------------------|
> | Rotary Embeddings      | 7025 (1.00x)             | 16908 (1.00x)             |
> | Cosine Embeddings q,k  | 10276 (1.46x)            | 28467 (1.68x)             |
> | Cosine Embeddings      | 10438 (1.49x)            | 29630 (1.75x)             |
>
>
> Comparison of training and inference speeds (in sequences per seconds) on the LRA’s Pathfinder task. Cosine RelPE are significantly faster in both scenarios. “q, k” in the second row denotes that Cosine RelPE were applied separately to Q and K matrices like in regular RoPE.
>
> Regarding the use of local attention, we discuss the motivation in the section 3.3 of the paper, to quote:
>
> > “The reason of this extension is outlined by Qin et al. [9]: in linear Transformer family of models, attention scores of a query are distributed along the sequence length more uniformly as compared to Softmax attention, so the model is not fully able to focus at details in the vicinity of a query’s token.”
>
> To reiterate this reasoning: intuitively, with local attention scheme we introduce a proximity bias which helps the model to pay more attention to close tokens in case of very large contexts. Standard self-attention achieves this property due to softmax nonlinearity which is able to selectively increase some of the attention scores by a large magnitude in relation to others.
>
> ---
>
> **References**
>
> [1] Katharopoulos et al., "Transformers are RNNs: Fast Autoregressive Transformers with Linear Attention." ICML 2020
>
>
> [2] Choromanski et al., "Rethinking Attention with Performers." ICLR 2021
>
> [3] Su et al., "RoFormer: Enhanced Transformer with Rotary Position Embedding." arXiv preprint arXiv:2104.09864, 2021
>
> [4] Biderman et al., "Pythia: A Suite for Analyzing Large Language Models Across Training and Scaling." ICML 2023
>
> [5] Black et al., "GPT-NeoX-20B: An Open-Source Autoregressive Language Model." BigScience Workshop 2022
>
> [6] Chowdhery et al., "PaLM: Scaling Language Modeling with Pathways." JMLR 2023
>
> [7] Dubey et al., "The Llama 3 Herd of Models." arXiv preprint arXiv:2407.21783, 2024.
>
> [8] Amos et al., "Never Train from Scratch: Fair Comparison of Long-Sequence Models Requires Data-Driven Priors." ICLR 2024
>
> [9] Qin et al., "The Devil in Linear Transformer." EMNLP 2022

---

> ### Author Response · Authors · 2024-11-27
> **General response. Part 3**
>
> **Weakness 4**
>
> > The results in Long Range Arena are promising insofar as they match up against a standard transformer and an SSM, but this may not be a fair comparison. Given that the authors start with a regular transformer and apply modifications to show improvement on long-context, they could also compare against more recent models specialized for long context.
>
> We thank you for the good suggestion. Indeed, Table 1 in the paper lists only two baselines. However, these baselines are exceptionally strong. The first baseline, the transformer you mentioned, is not rather standard but augmented with RoPE, which let it surpass all previous Transformer variants. And the paper, which introduced it, “Never Train from Scratch”, was published in ICLR recently, in 2024.
>
> Although it may be only subtly hinted at in the initial version of the manuscript (lines 424-425), by achieving superior results than our baseline from “Never Train from Scratch”, we automatically outperform *all other Transformer-based models and architectures*, which we are aware of, including the most recent ones, such as [1-2].
>
> Inspired by your and other reviewers’ suggestions, we composed a full comparison table, listing results for 25+ models. We share it in the General Response and will add it to the Appendix of the paper.
>
> > For example, the authors omit comparison with S5, whose numbers are publicly available on PapersWithCode, as well as S4 V2 and a long list of other models benchmarked on Long Range Arena but not necessarily added there.
>
> S4 [3], S5 [4], Megalodon [5] (current first place) and other models which occupy the top spots of the LRA suite of benchmarks belong to the class of State Space Models (SSMs).
>
> Generally, the SSMs are in the league of their own in terms of the modeling quality on the LRA, and until now, no Transformer-based model could close the gap on the LRA even to the relatively old SSM-based architectures. This insurmountable gap is explained by SSMs’ inherent inductive bias towards capturing hierarchical and long-range dependencies and lack of such bias in Transformers, as discussed in [1, 6-7].
>
> Since the difference in performance between Transformers and SSMs is very big, it is customary for the recent papers published in the leading ML venues not to provide comparisons with any SSMs on the LRA if they propose a new Transformer-based architecture and test it on this suite of benchmarks (see, e.g. [2] – ICLR 2024, [8] – ICLR 2022, and [9] – ICML 2023).
>
> In light of aforementioned arguments, we would like to emphasize that outperformance by DenseAttention of even relatively old S4-v1 SSM is a valuable and remarkable achievement for the Transformer-based architectures. This is the first case when such architecture surpasses an SSM on the 4 of 6 LRA benchmarks.
>
>
> **References**
>
> [1] Amos et al., "Never Train from Scratch: Fair Comparison of Long-Sequence Models Requires Data-Driven Priors." ICLR 2024
>
> [2] Zhang et al., "The Hedgehog & the Porcupine: Expressive Linear Attentions with Softmax Mimicry." ICLR 2024
>
> [3] Gu et al., "Efficiently Modeling Long Sequences with Structured State Spaces." ICLR 2022
>
> [4] Smith et al., "Simplified State Space Layers for Sequence Modeling." ICLR 2023
>
> [5] Ma et al., "Megalodon: Efficient LLM Pretraining and Inference with Unlimited Context Length." NeurIPS 2024
>
> [6] Ma et al., "Mega: Moving Average Equipped Gated Attention." ICLR 2023
>
> [7] Tran et al., "The Importance of Being Recurrent for Modeling Hierarchical Structure." EMNLP 2018
>
> [8] Qin et al., "cosFormer: Rethinking Softmax in Attention." ICLR 2022
>
> [9] Zandieh et al., "KDEformer: Accelerating Transformers via Kernel Density Estimation." ICML 2023

---

> ### Author Response · Authors · 2024-11-27
> **Author response. Part 4**
>
> **Weakness 5**
>
> > The result on efficiency compared to BERT also may seem to not be a fair comparison. BERT is trained with an encoder-only architecture, while DenseAttention Network is trained with a decoder-only architecture. A fairer comparison would pit DenseAttention Network against a regular decoder-only transformer (as well as BERT if desired, along with, say, an SSM), under the same experimental setting, and allow readers to observe trends in the different approaches as different scaling parameters vary.
>
> We apologize if some parts of the paper may have caused your confusion. However, we explicitly stated in the paper (section 4.2, lines 459-460 in the original manuscript):
> > “We pre-train an encoder model with the approximately same number of parameters as in BERT-large”
>
> We reiterate here that DenseAttention-BERT architecture is the bidirectional encoder-only and very closely follows the original architecture of [1]. In fact, we replicate the implementation details and training process for the original model as closely as possible, except for replacing Transformer blocks with DANet blocks. Our key goal was to show that DenseAttention-BERT is at least on par with the original model in terms of LM quality, while being faster and more computationally efficient, and we successfully accomplished this goal.
>
> Also, we have conducted additional experiments on scaling laws for the DenseAttention-BERT architecture and present them below.
>
> | Model              | Parameters | Configuration   | MLM loss | MLM accuracy, % |
> |--------------------|------------|------------------|----------|-----------------|
> | DANet-BERT-small   | 31M        | L=6, D=512      | 2.74     | 49.5           |
> | DANet-BERT-base    | 110M       | L=16, D=768     | 2.02     | 60.0           |
> | DANet-BERT-large   | 336M       | L=32, D=1024    | 1.70     | 64.9           |
>
>
> The table depicts three single-head DenseAttention Network models of different sizes pre-trained on Wiki+BookCorpus dataset with MLM objective for 100B tokens. MLM loss and accuracy are reported for out-of-sample data from C4 dataset [2]. L and D parameters denote number of layers and hidden dimension of FFN input, respectively.
>
> [1]  Devlin et al., "BERT: Pre-training of Deep Bidirectional Transformers for Language Understanding." NAACL 2019
> [2] Raffel et al., "Exploring the Limits of Transfer Learning with a Unified Text-to-Text Transformer." JMLR 2020
>
> ---
>
> **Question 1**
>
> > What is the specific motivation of designing CosineRelPE?
>
> Please see discussion for weakness 3.
>
> ---
>
> **Question 2**
> > Is there anything that suggests that any new block (DenseAttention, CosineRelPE, MaxNormActivation) can generalize to other architectures and improve either performance or efficiency (while not degrading the other)?
>
> Thank you for the inspiring question! Let us respond point-by-point:
>
> **1.** **DenseAttention Network** is a general architecture block which can serve as a drop-in replacement for the Transformer block in every model architecture that uses it. We conducted experiments on the diverse modalities spanning from logic and reasoning to language modeling and image classification, which are components of the LRA suite of benchmarks, along with standalone LM pre-training.
>
> Their results show that DenseAttention is capable of generalizing to many different tasks and achieving favorable performance in comparison with standard Transformer and its augmented variants while being faster and more computationally efficient. We believe it also strongly indicates DenseAttention can be ported or applied to specialized architectures that would benefit from long-context efficiency improvements, such as ViT [1] and SAM [2] for Computer Vision tasks and LLAMA [3] for language modeling. We actively plan to address them in future work.
>
> **2.** **Cosine RelPE** is a general building block which can be applied to any Transformer-like architecture in place of RoPE to bring efficiency gains. Based on both theoretical and empirical observations, we believe Cosine RelPE would contribute similarly favorably to these architectures in terms of modeling quality and are confident it will increase computational efficiency. However, a careful exploration of a general class of trigonometric relative position transformations and various methods of their application to attention inputs  should be performed, which is worth a dedicated research paper. We also leave it to future work.
>
> **3.** **Local Attention pattern** similar to ours (combination of alternating local and global attention layers) is already shown to perform well in Google’s Gemma 2 family of models [4].
>
> **4.** **MaxNormActivation** was designed to overcome the specific challenges of DenseAttention. It may prove to be useful in training deep NN models with large dimension of hidden states in fp16 format to prevent numerical instabilities.

---

> > ### Author Response · Authors · 2024-11-27
> > **References**
> >
> > [1] Dosovitskiy et al., "An Image is Worth 16x16 Words: Transformers for Image Recognition at Scale." ICLR 2021
> >
> > [2] Kirillov et al., "Segment Anything." ICCV 2023
> >
> > [3] Dubey et al., "The Llama 3 Herd of Models." arXiv preprint arXiv:2407.21783, 2024
> >
> > [4] Gemma Team, "Gemma 2: Improving Open Language Models at a Practical Size." arXiv preprint arXiv:2408.00118, 2024

---

> ### Comment · Reviewer_rVwA · 2024-11-27
> **Response to Rebuttal**
>
> Thanks for the detailed response.
>
> > Framing the absence of any transformations applied to $Q$ and $K$ as an identity mapping $\phi(x) = x$ would be conceptually wrong because transformed values are required to be non-negative in the Linear Transformer class of algorithms [1-2]. And the absence of reweighting the attention scores by their row-wise sums further sets DenseAttention apart from Linear Transformers.
>
> Thanks for pointing it out. This seems to make sense --- actually $\phi$ need not be non-negative (as a scalar or element-wise as a vector) but $\phi(x)^{\top}\phi(y)$ needs to be non-negative (to clarify a point being made in your general response), for example Gaussian kernel. But overall I agree that this dense attention module is not strictly in scope of [1].
>
> > Furthermore, calculation of $\|x\|_{2}$ in low precision formats can lead to either a numerical instability (as in the case of fp16) or to a loss of numerical precision (both in fp16 and bf16) for high-dimensional $x$.
>
> This makes sense and is a good motivation for MaxNormActivation.
>
> However, the CosineRelPE block, which is the other novel block, still seems to only be motivated by an ablation and concerns of efficiency. In order to validate that it is a full replacement of RoPE without any performance drawbacks, it would have been great to have higher-scale experiments. Local attention is pulled from another work so it is OK to not have the full motivation for it here.
>
> > “We pre-train an encoder model with the approximately same number of parameters as in BERT-large”
>
> My apologies, I seem to have missed the word "encoder" in my initial review. Yes, this experiment makes sense now, and seems to be fair.
>
> > Question 2
>
> The response to this question contains mostly a summary of the tools used. I would have liked to see if there are any fundamental reasons or empirical proof why the blocks can or cannot generalize beyond the context of having all three modifications together. As in, I understand that you can arbitrary replace the standard transformer blocks with them; my question is more about understanding the contexts in which they work. For example, can the DenseAttention block work with standard LayerNorms (I suspect the answer is no for the reasons discussed earlier in your rebuttal)? Can the MaxNormActivation work without DenseAttention, i.e., does it improve performance or efficiency of transformers with the usual attention? The ablation on CosineRelPE seems to have already been tried in Table 2.
>
> Due to comprehensively addressing several of my comments in the initial review, I will raise my score.

---

> > ### Author Response · Authors · 2024-11-29
> > **Thanks and further discussion**
> >
> > Dear reviewer rVwA,
> >
> > We thank you very much for your response and appreciation of our work. We are delighted to have partially addressed your concerns.
> >
> > > In order to validate that it is a full replacement of RoPE without any performance drawbacks, it would have been great to have higher-scale experiments.
> >
> > In our preliminary ablations we have found that incorporation of either RoPE or Cosine RelPE causes a performance boost on all of the LRA tasks. One of such ablations have been presented in Section 4.1, Table 2 in the paper (we repeat below the relevant part for your convenience)
> >
> >
> > | **Model**                                         | **Accuracy** |
> > |---------------------------------------------------|--------------|
> > | DANet + Sinusoidal Embedding (bf16 format)        | 82.69        |
> > | DANet + Cosine RelPE                              | 83.98        |
> >
> > Ablation on the type of embeddings on the Retrieval task of the LRA. Use of Cosine RelPE leads to a better performance. We observe a similar effect on all other LRA tasks.
> >
> > As the difference in metrics of models trained on either of PE variants had been negligible, and at the same time, a boost from using them in general was evident, we discarded the full logs and proceeded with the most computationally efficient option (Cosine RelPE) as the default one. Should you recommend it, we will reproduce these full comparisons and include them in the final version of the paper.
> >
> > > For example, can the DenseAttention block work with standard LayerNorms (I suspect the answer is no for the reasons discussed earlier in your rebuttal)?
> >
> > You are correct. Indeed, it proved empirically that DenseAttention can’t work with a standard LayerNorm. As we explicitly state in the paper (lines 255-256), to quote: “In our ablation experiments any other activation or normalization function or absence thereof would lead to a prompt and unrecoverable numerical instability early on during training.”
> >
> > > Can the MaxNormActivation work without DenseAttention, i.e., does it improve performance or efficiency of transformers with the usual attention?
> >
> > Thank you for the suggestion! We are now actively working on this experiment and will do our best to finish and present it before the discussion period ends.
> >
> > ---
> >
> > Again, we are grateful for your feedback and subsequent response as they helped to clarify and enhance our work, and we are looking forward to further discussion if you have any suggestions or remaining concerns.

---

> ### Author Response · Authors · 2024-12-02
> **Additional experiment for reviewer rVwA**
>
> Dear reviewer rVwA,
>
> Following your suggestion and fulfilling the commitment we expressed in the latest response, we have completed an additional experiment. We present its results below and will include in the final version of the manuscript:
>
> | **Model** | **MLM loss** | **MLM accuracy** |
> |-------------------------------------|--------------|-------------------|
> | BERT-large (LayerNorm) | 2.11 | 59.3 |
> | BERT-large (MaxNormActivation) | 2.46 | 54.3 |
>
> Comparisons between LayerNorm and MaxNormActivation for BERT-large Transformer pre-trained on Wiki+BookCorpus dataset for 10B tokens. MLM loss and accuracy are reported for out-of-sample data from C4 dataset
>
> The results indicate that standard LayerNorm is optimal for standard Transformer as replacing it by MaxNormActivation leads to a subpar performance. On the other hand, MaxNormActivation is not a merely optimal but rather essential part of DANet architecture, because putting the standard LayerNorm into it instead of MaxNormActivation results in numerical instability.
>
> ---
>
> We thank you for the opportunity to further enhance our work and are keen to receive your feedback.

---

> ### Author Response · Authors · 2024-12-03
> **Gentle follow up for reviewer rVwA**
>
> Dear reviewer rVwA,
>
> We express heartfelt gratitude for your response and constructive, actionable feedback which resulted in an additional experiment and extended discussion. As we hope to have resolved your remaining concerns, and given that the discussion period ends in less than 12 hours, we kindly ask you to consider updating your score if we have been able to address them, or to let us know if there are any remaining concerns or questions.

---

### Official Review · Reviewer_oGkz · 2024-10-29

**Soundness:** 2
**Presentation:** 2
**Contribution:** 1
**Rating:** 5
**Confidence:** 4

**Summary:**

This paper proposes a novel neural network architecture called DenseAttention Network as alternative to Transformer networks with Self-Attention.
The core innovation of DANet is the novel DenseAttention mechanism, which removes Softmax and projection layers from original Self-Attention.
Additionally the authors modify the surrounding network block: They replace Layernorm or RMS with their novel “MaxNormActivation”, they remove some skip connections and modify the Rotary Positional Embeddings.
The paper performs experiments on the Long Range Arena Benchmark and masked language modeling with BERT-large sized models.

The paper claims to outperform a BERT baseline on masked language modeling.
It claims to set a new SOTA on “Transformer-based” models on LRA and to outperform 4 of 6 State Space model baselines.

**Strengths:**

- The usage of MaxNormActivation seems to be well motivated by a theoretical variance analysis. However, this could also be supported with experiments empirically too.
- Code provided.

**Weaknesses:**

In general I believe this paper is not ready for publication as there are several weaknesses in terms of the new architecture, the experiments and the presentation in the paper. My main concerns are summarized below:

- A related work section is missing in which the authors put DANet in relation to other Linear Attention variants (e.g. GLA https://arxiv.org/abs/2312.06635 ), State space models (e.g. Mamba https://arxiv.org/abs/2405.21060) or other RNNs variants (e.g. xLSTM (https://arxiv.org/abs/2405.04517 ) or RWKV (https://arxiv.org/abs/2305.13048 )). Also a relation to embedding models other than BERT is missing, e.g. Monarch Mixer (https://arxiv.org/abs/2310.12109).
- Since DANet seems to be a hybrid architecture (Section 3.3), also a relation to hybrid architectures (e.g. https://arxiv.org/abs/2402.19427, https://arxiv.org/abs/2406.07522) is interesting.
- There are so many architecture changes (e.g. Layernorm, Positional Encoding, Block structure, Attention mechanism, Block order / hybrid variants) that leave the reader unclear of what brings performance gains. A careful ablation study could help here.
- While the paper demonstrates large throughput benefits in the long context regime compared to Transformers, it has not been shown in the paper that DANet performs well in the long context regime.
- Regarding Cosine RelPE: It is not clear why the authors made the modification to the original Rotary Positional Embedding. It seems to be motivated by efficiency gains, but this claim is not supported sufficiently. An experiment on this could help.
- A conclusion is missing.

**Questions:**

- Why do you still consider DANet as Transformer-based? The only part of transformers that is left, is the Feeforward layers which is now inside the block.
- You train your model with 4 stages, but the original BERT was trained on 2 stages. Could you also train the baseline in the same way?
- On page 10, line 494 (key highlights) you hint at the fact that DANet outperforms the baseline due to a soft-capping of output logits that you use. Why did you not try this for the baseline as well?
- L.400: The authors find that local attention is effective. Do you use the Transformer Self-Attention here? An ablation on this would be interesting.
- Why do you use float16 in the experiments?

---

> ### Author Response · Authors · 2024-11-29
> **Response to Reviewer oGkz. Part 1**
>
> We thank the reviewer oGkz for the thoughtful feedback, attention to the details, and constructive suggestions which have helped us to improve the exposition of our work and prompted us to conduct several additional experiments. We are grateful to you for the comprehensive outline of our contributions and recognition of motivation behind MaxNormActivation.
>
> Please let us try to address and alleviate your concerns with detailed responses.
>
> ---
>
> **Question 1**
> > Why do you still consider DANet as Transformer-based? The only part of transformers that is left, is the Feeforward layers which is now inside the block.
>
> Please let us start by answering the first question to set the context for further discussion.
>
> The defining part of the standard Transformer architecture is its attention mechanism as all other constituents may be altered, replaced (e.g standard FeedForward for GeGLU/ SwiGLU layers and LayerNorm for RMSNorm like in [1]), or moved (PreLayerNorm and PostLayerNorm discussed in [2])
>
> Fundamentally, the attention mechanism A can be described as follows:
>
> $\mathbf{A} = \mathcal{S} ( \mathbf{Q}(\mathbf{X}) \mathbf{K}^\top (\mathbf{X}) ) \mathbf{V} (\mathbf{X})$
>
> where $\mathbf{Q}(\cdot) \in \mathbb{R}^{N \times d_q}$, $\mathbf{K}(\cdot) \in \mathbb{R}^{N \times d_q}$ and  $\mathbf{V}(\cdot) \in \mathbb{R}^{N \times d_v}$ are some mappings (usually, linear projections) of $\mathbf{X} \in \mathbb{R}^{N \times d}$, and $\mathcal{S}(\cdot)$ is some similarity function (for example, row-wise $\text{Softmax}(\cdot \/ d_q)$ in the standard Transformer’s attention).
>
> All Transformer-based architectures have an attention mechanism which can be parametrized by some of functions $\mathbf{Q}(\cdot)$, $\mathbf{K}(\cdot)$, $\mathbf{V}(\cdot)$, and $\mathcal{S}(\cdot)$. In particular, by letting $\mathbf{Q}(\mathbf{X}) = \mathbf{X} \mathbf{W}_Q$, $\mathbf{K}(\mathbf{X}) = \mathbf{X}$, $\mathbf{V}(\mathbf{X})=\mathbf{X}$, and $\mathcal{S}(\mathcal{X}) = \mathcal{X}$, we get
>
> $\mathbf{A} = \mathbf{Q} \mathbf{K}^\top \mathbf{V} = \mathbf{X} \mathbf{W}_Q \mathbf{X}^\top \mathbf{X},$
>
> which is exactly the formula for DenseAttention.
>
> In contrast, DenseAttention cannot be considered an instance of another broad class of algorithms – State Space Models (SSMs) / linear RNNs which are characterized by linear recurrence:
>
>  \begin{align*}
> &x_t = \mathbf{\overline{A}}x_{t-1}  + \mathbf{\overline{B}}u_t \\
> & y_t = \mathbf{\overline{C}}x_t + \mathbf{\overline{D}}u_t,
> \end{align*}
>
> where $\mathbf{\overline{A}}$ is a data-independent matrix. There are no data-independent matrices in DenseAttention.
>
> Therefore, DANet is most naturally classified as an instance of the Transformer-based class of algorithms. In fact, we specifically designed it by taking the standard Transformer architecture and simplifying / modifying it.
>
> ---
>
> **Weakness 1**
> > A related work section is missing in which the authors put DANet in relation to other Linear Attention variants (e.g. GLA https://arxiv.org/abs/2312.06635 ), State space models (e.g. Mamba https://arxiv.org/abs/2405.21060) or other RNNs variants (e.g. xLSTM (https://arxiv.org/abs/2405.04517 ) or RWKV (https://arxiv.org/abs/2305.13048 )). Also a relation to embedding models other than BERT is missing, e.g. Monarch Mixer (https://arxiv.org/abs/2310.12109).
>
> We thank you for pointing out limited discussions with other work. In fact, we cited several papers on linear and sub-quadratic Transformers, including one of the first and most influential works on Linear Attention [3] in the Introduction (lines 52-53), and LocalAttention for DenseAttention (lines 361-363; 373 where we discussed similarities with our local attention paradigm) sections.
>
> However, we initially omitted the extended discussion of [3] and subsequent work as we believe DenseAttention architecture is conceptually different from Linear Transformer and its derivatives. Instead, we briefly discussed the most similar research to ours [4-5] in the Introduction.
>
> Motivated by your and other reviewers’ feedback, we wrote a detailed exposition which puts DenseAttention into perspective with other Linear Attention variants. We present it in the General Response and gently ask you to read it.
>
> As discussed earlier, DenseAttention Network is a Transformer-based model and is not related to SSMs and RNNs. However, encouraged by your comment, we further augment this section by a brief exposition of these architectures. We present it below in full.
>
> ---

---

> ### Author Response · Authors · 2024-11-29
> **Response to Reviewer oGkz. Part 2**
>
> **Other sub-quadratic algorithms for sequence processing**
>
> Another promising line of work focuses on applying deep State Space Models (SSMs) [6-9] and Linear RNNs [10-12] to long-range sequence and language modeling. Fundamentally, these architectures model interactions in sequence dimension by a linear recurrence:
>
> $
> \begin{align*}
> &x_t = \mathbf{\overline{A}}x_{t-1}  + \mathbf{\overline{B}}u_t, \\
> & y_t = \mathbf{\overline{C}}x_t + \mathbf{\overline{D}}u_t,
> \end{align*}
> $
>
> where recurrence matrix $\mathbf{\overline{A}}$ and other parameters are data-independent matrices which form and initialization are defining properties for a particular SSM/ RNN architecture.
> The linear recurrence is advantageous during inference as it runs in $O(N)$ time. For training, it also can be unrolled into a convolutional kernel $\mathbf{K} = \begin{bmatrix} \mathbf{\overline{C}  \overline{B}}, & \mathbf{\overline{C} \overline{A} \overline{B}}, & \ldots, & \mathbf{\overline{C} \overline{A}}^{N-1}\mathbf{\overline{B}} \end{bmatrix} $ to compute $y = \mathbf{K} * u$ via Fast Fourier Transform (FFT) in $O (N \log N)$ time. Here, we set $\mathbf{D}=0$ for ease of exposition, but in practice it's usually set to identity to act as a skip-connection ubiquitous in modern deep NN architectures.
>
> Among other novel algorithms which rely on FFT or its generalizations such as Monarch matrices [13], are Long Convolutions [14], Hyena [15], and Monarch Mixer [16], with the latter using sub-quadratic primitives both for computations along the sequence length and the model dimension.
>
> While being sub-quadratic, these algorithms are still slower than linear time as in DenseAttention. However, recently [9] introduced data-dependent gating for SSM parameters and low-level, hardware efficient CUDA implementation for parallel-scan operation which allows for fast linear-time processing both during training and inference. And [17] adopt a similar gating mechanism for causal linear attention which allows to drop the denominator in the linear attention formula but also admits no parallel training without resorting to low-level implementations.
>
> ---
>
> References:
>
> [1] Touvron et al., "LLaMA: Open and Efficient Foundation Language Models." arXiv preprint arXiv:2302.13971, 2023.
>
> [2] Xiong et al., "On Layer Normalization in the Transformer Architecture." ICML 2020
>
> [3] Katharopoulos et al., "Transformers are RNNs: Fast Autoregressive Transformers with Linear Attention." ICML 2020
>
> [4] Shen et al., "Efficient Attention: Attention with Linear Complexities." WACV 2021
>
> [5] Koohpayegani and Pirsiavash, "SimA: Simple Softmax-free Attention for Vision Transformers." WACV 2024
>
> [6] Gu et al., "Efficiently Modeling Long Sequences with Structured State Spaces." ICLR 2022
>
> [7] Gupta et al., "Diagonal State Spaces are as Effective as Structured State Spaces." NeurIPS 2022
>
> [8] Ma et al., "Mega: Moving Average Equipped Gated Attention." ICLR 2023
>
> [9] Gu and Dao, "Mamba: Linear-Time Sequence Modeling with Selective State Spaces." CoLM 2024
>
> [10] Beck et al., "xLSTM: Extended Long Short-Term Memory." NeurIPS 2024
>
> [11] Orvieto et al., "Resurrecting Recurrent Neural Networks for Long Sequences." ICML 2023
>
> [12] Peng et al., "RWKV: Reinventing RNNs for the Transformer Era." Findings of EMNLP 2023
>
> [13] Dao et al., "Monarch: Expressive Structured Matrices for Efficient and Accurate Training." ICML 2022
>
> [14] Fu et al., "Simple Hardware-Efficient Long Convolutions for Sequence Modeling." ICML 2023
>
> [15] Poli et al., "Hyena Hierarchy: Towards Larger Convolutional Language Models." ICML 2023
>
> [16] Fu et al., "Monarch Mixer: A Simple Sub-Quadratic GEMM-Based Architecture." NeurIPS 2023
>
> [17] Yang et al., "Gated Linear Attention Transformers with Hardware-Efficient Training." ICML 2024
>
> ---
> We thank you for the opportunity to improve our paper and will include the above exposition in the next revision of the manuscript (please note that the present revision contains an abbreviated un-proofread version which will be updated).

---

> ### Author Response · Authors · 2024-11-29
> **Response to Reviewer oGkz. Part 3**
>
> **Weakness 2 & Question 4**
>
> > Since DANet seems to be a hybrid architecture (Section 3.3), also a relation to hybrid architectures (e.g. https://arxiv.org/abs/2402.19427, https://arxiv.org/abs/2406.07522) is interesting.
>
> > L.400: The authors find that local attention is effective. Do you use the Transformer Self-Attention here?
>
> We apologize if some parts of the paper may have caused your confusion. Our LocalAttention and ShiftedLocalAttention layers are just regular DenseAttention layers applied to non-overlapping chunks of a sequence rather than globally, similar to [1-2]. Although it was implied in text and shown in the code, we will state it explicitly in the next version of the paper to make it clear.
>
> We use only DenseAttention and no Transformer’s Self-Attention in our local attention scheme which alternates local and global layers. Thus, DANet augmented with local attention is still a “pure” architecture and cannot be directly related to hybrid Transformer-SSM models mentioned in your comment.
>
> However, as we referenced in the Section 3.3, there is  a “pure” Transformer Large Language Model which successfully employs a similar alternated pattern at the scale of billions of parameters – Gemma 2 by Google.
>
> > An ablation on this would be interesting.
>
> In fact, we conducted two ablation studies on the effectiveness of local attention. The first is available in the main text of the original and current revisions (Section 4.1, Table 2). It shows that inclusion of local attention significantly boosts modeling quality on the LRA. A new ablation, centered around long contexts in DenseAttention-BERT, will be reported in response to Weakness 4 (*DANet-BERT 16K with local attention ablation*).
>
> References:
>
> [1] Dao et al., "FlashAttention: Fast and Memory-Efficient Exact Attention with IO-Awareness." NeurIPS 2022
>
> [2] Qin et al., "The Devil in Linear Transformer." EMNLP 2022
>
> [3] Gemma Team, "Gemma 2: Improving Open Language Models at a Practical Size." arXiv preprint arXiv:2408.00118, 2024.
>
> ---
>
> **Weakness 3**
>
> > There are so many architecture changes (e.g. Layernorm, Positional Encoding, Block structure, Attention mechanism, Block order / hybrid variants) that leave the reader unclear of what brings performance gains. A careful ablation study could help here.
>
> We thank you for bringing up this very interesting topic and are happy to address it with a detailed explanation.
>
> The bottom line is that while we were able to conduct some of them, not all ablations are possible in principle.
>
> **Attention Mechanism, LayerNorm & Block Structure.** One of the core contributions of our work is the complete removal of Softmax. As discussed in the paper (lines 215-226), without it attention outputs become unbounded and quickly diverge to $\infty$ or shrink to 0, especially when computed in half-precision format fp16.  The only way we found to prevent it was to use MaxNormActivation, which we had derived by theoretical analysis, before the attention layer. As we explicitly stated in the paper (lines 255-257), to quote:
>
> > In our ablation experiments any other activation or normalization function or absence thereof would lead to a prompt and unrecoverable numerical instability early on during training.
>
> The cubic growth rate of outputs w.r.t inputs in DenseAttention dictated another design choice: moving the second MaxNormActivation to the end of the DANet block after FFN sub-block. As the standard Transformer block has only two LayerNorms, we aimed to keep this number intact. Leaving any type of a layer normalization between attention and FFN sub-blocks instead of placing it in the end would cause the loss to diverge or get trapped in a bad local minimum for moderately deep networks.
>
> To summarize, the absence of Softmax does not leave much room for architectural changes and, thus, ablations, as the current choice and placement of the components make it numerically stable. However, we conducted a study on a parameter which allowed for an ablation: the number of heads. We presented it in Section 4.2, Table 3. The results indicate that the single head DANet-BERT variant  mostly outperforms the multi-head one (extended analysis is available in lines 496-501).
>
> **Block Order / Hybrid Variants** We discuss and present the ablations for the Local Attention in the responses to Weakness 2 and Weakness 4 (*DANet-BERT 16K with local attention ablation*).
>
>  **Positional Encoding** We present the ablations for Cosine RelPE in the response to Weakness 5.

---

> ### Author Response · Authors · 2024-11-29
> **Response to Reviewer oGkz. Part 4**
>
> **Weakness 4**
>
> > While the paper demonstrates large throughput benefits in the long context regime compared to Transformers, it has not been shown in the paper that DANet performs well in the long context regime.
>
> ---
>
> **The LRA results**
>
> Our primary experiments involved testing on the Long Range Arena suite of benchmarks [1]. It’s a diverse and challenging set of tasks designed *specifically to stress test the modeling performance of an architecture in the long context regime*. The sequence length for the tasks varies from 1K (smallest) to 16K tokens. Until recently, no Transformer-based model could even score above chance on the most challenging benchmark, Path-X with 16k context length.
>
> Prompted by your comments, we wrote up a detailed description of the LRA tasks. We present it in the Appendix E.1 in the current revision and politely encourage you to read it but omit posting it in this comment for brevity.
>
> We would like to emphasize that the LRA is arguably considered to be a gold standard in testing long-range abilities of NN sequence models. And DenseAttention architecture outperforms, to the best of our knowledge, all other Transformer-based architecture which have been tested on the LRA to date.
>
> Although it may be only subtly hinted at in the manuscript (lines 430-431), by achieving superior results than our exceptionally strong baseline from “Never Train from Scratch” [1], we automatically outperform all other Transformer-based models and architectures, including the most recent ones, such as [1-2].
>
> Inspired by your and other reviewers’ comments, we composed a full comparison table, listing results for 25+ models. We share it in the General Response and in the Appendix E.2 in the current revision of the paper.
>
> Moreover, our second main baseline for the LRA – S4 model [3] – is a State Space Model.  Generally, SSMs are in the league of their own in comparison with Transformer-based architectures and greatly outperform them on the LRA benchmarks due to their inherent inductive bias towards capturing hierarchical and long-range dependencies and lack of such bias in Transformers, as discussed in [1, 4-5]. DenseAttention outperforms this much stronger SSM baseline in 4 of 6 benchmarks (full results reported in Section 4.1, Table 1 in the paper). To the best of our knowledge, this is the first case when a pure Transformer-based model compares favorably with an SSM on the LRA, which indicates its potent long-range modeling abilities.
>
> ---
>
> **Pathfinder-256 benchmark**
>
> Furthermore, motivated by your comments, we conducted an experiment on Pathfinder-256 benchmark. It is is an extremely challenging version of the Pathfinder task with sequence length 65k which is on par with input context size of even recent generations of open-source  (e.g. DBRX, 32k, https://www.databricks.com/blog/introducing-dbrx-new-state-art-open-llm) and close source (original GPT-4, 32k) Large Language Models.
>
> We present the results and the discussion below in full (It’s also available in the Appenix H.1 in the revised paper).
>
> ---
>
> | Algorithm                                | Accuracy on the validation set, % |
> |-----------------------------------------|------------------------------------|
> | FlashAttention [6]                      | 63.1                               |
> | S4 [3]                                 | 67.8                               |
> | DenseAttention                          | _72.6_                               |
> | DenseAttention after additional 550 epochs | **77.1**                            |
>
>
> DenseAttention model outperforms existing results from the literature of standard Transformer augmented with FlashAttention [9] and S4-v2 model [12] as reported in [11] The result holds both when the training procedure is carried out for 200 training epochs as in [9] and then it’s prolonged for 550 additional epochs.
>
> This experiment lets us make several observations:
>
> * DenseAttention Network architecture performs well even on very long input sequences which is promising given current trend of increasing context size in modern Large Language and Multimodal Models
> * DenseAttention shows favorable scaling properties with respect to the amount of training iterations, even with the fixed dataset size. The validation accuracy for the task kept improving throughout the whole training and would likely have continued if the experiment had not been stopped.
> * Truly linear scaling in sequence length is crucial for improvements in quality for large contexts. It took approximately 3 days on 4 H100 GPUs to train our model for 750 epochs in linear mode, while the projected runtime of quadratic (FlashAttention-2 [10]) and log-linear (S4) algorithms in the same setting would be at best 3 and 0.5 months, respectively, which renders them impractical for prolonged training.
> ---

---

> ### Author Response · Authors · 2024-11-29
> **Response to Reviewer oGkz. Part 5**
>
> **DANet-BERT 16K with local attention ablation**
>
> Finally, we performed new experiments by taking DANet-BERT model after it had finished pre-training on sequence length 512, augmenting it with our local attention scheme and continuing pre-training on sequence lengths 1024 and then 16384 tokens. We compare the results with the old way of pre-training without local attention and report them in full below (as we obtained them recently, they are not yet present in the current revision of manuscript).
>
> | **Context size**      | **1k**                    |                           |                           | **16k**                    |                           |                           |
> |------------------------|---------------------------|---------------------------|---------------------------|----------------------------|---------------------------|---------------------------|
> | **Metrics**           | **Samples**              | **MLM loss**              | **MLM acc.**              | **Samples**               | **MLM loss**              | **MLM acc.**              |
> | DANet-BERT         | 80M                      | 2.255                     | 0.591                     | 27M                       | 2.843                     | 0.452                     |
> | DANet-BERT + local attention | 80M            | 1.705                     | 0.647                     | 7.8M                      | 1.689                     | 0.637                     |
>
>
> Comparison of DenseAttention BERT-large pre-trained on long context sizes with and without local attention. The models with context size 1k and 16k were evaluated on the corresponding length texts from C4 and Bookcorpus (held-out split) datasets respectively. Samples denotes number of sequences of corresponding length seen by a model during continual pre-training.
>
> The results show that introduction of local-global attention pattern helps to quickly recover the modeling performance even on extremely long sequences. It brings the performance to the same level we observed when pre-training on small sequences and significantly outperforms the models which were pre-trained without the local attention.
>
> ---
>
> To conclude, all of the results mentioned above indicate towards strong long context modeling capabilities of DANet architecture.
>
> ---
>
> **References:**
>
> [1] Amos et al., "Never Train from Scratch: Fair Comparison of Long-Sequence Models Requires Data-Driven Priors." ICLR 2024
>
> [2] Zhang et al., "The Hedgehog & the Porcupine: Expressive Linear Attentions with Softmax Mimicry." ICLR 2024
>
> [3] Gu et al., "Efficiently Modeling Long Sequences with Structured State Spaces." ICLR 2022
>
> [4] Ma et al., "Mega: Moving Average Equipped Gated Attention." ICLR 2023
>
> [5] Tran et al., "The Importance of Being Recurrent for Modeling Hierarchical Structure." EMNLP 2018
>
> [6] Dao et al., "FlashAttention: Fast and Memory-Efficient Exact Attention with IO-Awareness." NeurIPS 2022
>
> [7] Dao, T., "FlashAttention-2: Faster Attention with Better Parallelism and Work Partitioning." ICLR 2024
>
> **Weakness 5**
>
> > Regarding Cosine RelPE: It is not clear why the authors made the modification to the original Rotary Positional Embedding. It seems to be motivated by efficiency gains, but this claim is not supported sufficiently. An experiment on this could help.
>
> We thank you for the constructive feedback. Motivated by it, we conducted an additional ablation study, a discussion and results of which we present below (also available in the Appendix H.2 of the revised manuscript).
>
> ---
>
> **Ablation on Cosine RelPE**
>
> Regular Rotary Positional Embeddings (RoPE) [1] are known to enhance modeling performance and generalization in Transformer models and are widely used [2-5]. However, regular RoPE are not computationally efficient as we explain in section 3.2 of the paper. Our primary motivation behind designing Cosine RelPE is speed and efficiency gains, as we aimed to make DenseAttention as efficient as possible. As we demonstrated in the paper, expanded expressions for RoPE and Cosine RelPE are similar while the latter form of embeddings involves much less memory-intensive computations. Empirically, we found that the difference in modeling quality between the two types is negligible.
>
>
>
> | Model variant          | Training Speed (speed-up) | Inference Speed (speed-up) |
> |------------------------|---------------------------|----------------------------|
> | Rotary Embeddings      | 7025 (1.00x)             | 16908 (1.00x)             |
> | Cosine Embeddings q,k  | 10276 (1.46x)            | 28467 (1.68x)             |
> | Cosine Embeddings      | 10438 (1.49x)            | 29630 (1.75x)             |
>
> Comparison of training and inference speeds (in sequences per seconds) on the LRA’s Pathfinder task. Cosine RelPE are significantly faster in both scenarios. “q, k” in the second row denotes that Cosine RelPE were applied separately to Q and K matrices like in regular RoPE.

---

> ### Author Response · Authors · 2024-11-29
> **Response to Reviewer oGkz. Part 6**
>
> References:
>
> [1] Su et al., "RoFormer: Enhanced Transformer with Rotary Position Embedding." arXiv preprint arXiv:2104.09864, 2021
>
> [2] Biderman et al., "Pythia: A Suite for Analyzing Large Language Models Across Training and Scaling." ICML 2023
>
> [3] Black et al., "GPT-NeoX-20B: An Open-Source Autoregressive Language Model." BigScience Workshop 2022
>
> [4] Chowdhery et al., "PaLM: Scaling Language Modeling with Pathways." JMLR 2023
>
> [5] Dubey et al., "The Llama 3 Herd of Models." arXiv preprint arXiv:2407.21783, 2024.
>
> ---
>
> **Weakness 6**
>
> > A conclusion is missing.
>
> We thank you for pointing this out. We present it in the revised version of the paper and here, below in full.
>
> ---
>
> **Conclusion and Future Work**
>
> In this paper, we propose DenseAttention Network -- a general architecture which simplifies the Transformer block and can serve as a drop-in replacement in every model architecture using it. We conduct experiments on the diverse modalities spanning from logic to language modeling and image classification and from short to extremely long sequence lengths using the LRA suite of benchmarks and MLM-style language model pre-training on text data. The results show that DenseAttention is capable of generalizing to many different tasks and context sizes and achieving favorable performance in comparison with standard Transformer and its augmented variants while being faster and more computationally efficient even with no specialized, low-level computation algorithms such as in [1].
>
> We acknowledge that there are other modalities and specialized architectures that would benefit from long-context efficiency improvements if the DenseAttention is ported or applied to them, such as ViT [2] and SAM [3] for Computer Vision tasks, and LLAMA
> [4] for decoder-style language modeling. We hope to address them in future work. In particular, we look forward to adapting DenseAttention architecture to causal LLAMA-style LLMs and studying their scaling laws at billions of parameters range.
>
> References:
>
> [1] Dao et al., "FlashAttention: Fast and Memory-Efficient Exact Attention with IO-Awareness." NeurIPS 2022
>
> [2] Dosovitskiy et al., "An Image is Worth 16x16 Words: Transformers for Image Recognition at Scale." ICLR 2021
>
> [3] Kirillov et al., "Segment Anything." ICCV 2023
>
> [4] Touvron et al., "LLaMA: Open and Efficient Foundation Language Models." arXiv preprint arXiv:2302.13971, 2023.
>
> ---
>
> **Question 1**
>
> Addressed at the beginning of the response.
>
> ---
>
> **Questions 2 & 3**
>
> > You train your model with 4 stages, but the original BERT was trained on 2 stages. Could you also train the baseline in the same way?
>
> > On page 10, line 494 (key highlights) you hint at the fact that DANet outperforms the baseline due to a soft-capping of output logits that you use. Why did you not try this for the baseline as well?
>
> We thank you for the insightful questions. The first 2 of 4 stages corresponded to the same way the original BERT had been trained. Thus, after the second stage the DANet-BERT corresponds to fully pre-trained original BERT. The last two stages involve training on longer-context sequences (1k and 16k) to test the generalization abilities of our model. We emphasize that we report results for all 4 stages separately in Tables 3 and 6.
>
> The soft-capping of output logits and input embeddings had been a part of the DANet-BERT initially. However, motivated by your questions, we removed all the differences in the architecture of DANet-BERT from the original model to make them perfectly identical except for DANet blocks. Then we completely re-trained and re-evaluated the model. We present the new results in the table below and in the response to Weakness 4 (*DANet-BERT 16K with local attention ablation*) and we will include them into the next revision of the manuscript (as these experiments have finished only very recently, we haven’t been able to adjust the current revision).
>
> ---

---

> ### Author Response · Authors · 2024-11-29
> **Response to Reviewer oGkz. Part 7**
>
> **New results of DANet-BERT pre-training**
>
> | **Model**                            | **MLM Loss (L=128)** | **Acc. (L=128)** | **MLM Loss (L=512)** | **Acc. (L=512)** | **MLM Loss (L=1024)** | **Acc. (L=1024)** |
> |--------------------------------------|-----------------------|-------------------|-----------------------|-------------------|-----------------------|-------------------|
> | DenseAttention (1 head, N=128)       | **1.91**             | **0.620**         | -                    | -                 | -                     | -                 |
> | DenseAttention (4 heads, N=128)      | 1.97                 | 0.611             | -                    | -                 | -                     | -                 |
> | BERT-large                           | 2.58                 | 0.582             | 2.31                 | 0.614              | -                     | -                 |
> | DenseAttention (1 head, N=512)       | 1.97                 | 0.614             | 1.726                 | 0.644             | -                     | -                 |
> | DenseAttention (4 heads, N=512)      | 2.04                 | 0.602             | 1.84                 | 0.624             | -                     | -                 |
> | DenseAttention (1 head, N=1024)      | 1.96                 | 0.615             | **1.71**             | **0.648**         | 2.26                  | 0.591             |
> | DenseAttention (4 heads, N=1024)     | 2.07                  | 0.598             | 1.83                 | 0.627             | **1.87**             | **0.618**         |
>
> Evaluations of MLM loss and accuracy for DenseAttention Network models and the original BERT on C4 dataset texts of different context sizes. N is the maximum sequence length with which a model was trained or evaluated and L is the length of evaluation samples. DANet-BERT model variations uniformly outperform the original BERT on corresponding context sizes.
>
> The results of the new DANet-BERT model, which is closely aligned with the original, are similar with the old results. The relations and trends in performance between all models and all context lengths remain unchanged.
>
> ---
> **Question 5**
>
> >Why do you use float16 in the experiments?
>
> We thank you for this exciting question!
>
> We pay great attention to the compatibility of the DenseAttention with half-precision formats (fp16 and bf16) because they are widely, if not predominantly, used in practice, e.g for training and inference of Language Models. Importantly, use of these formats brings enormous speed gains (up to 2x in comparison with full precision)
>
> Speaking of the differences between the two formats, they have trade-offs in relation to each other. bf16 has a wider numerical range than fp16, but less precision which makes fp16 empirically better for some models provided that their activations stay inside its numerical range.
>
> Our primary motivation for using fp16 in experiments is its superior compatibility. A model, trained in fp16, can be converted and used with bf16, but the opposite need not always be the case. And a lot of hardware still used both in academia and in industry has no support for bf16 (e.g NVIDIA V100 and T4 server GPUs and Nvidia consumer GPUs older than 3xxx series). We would like to make the architecture and the models accessible to everyone so we try to use fp16 wherever possible.
>
> ---
>
> **P. S.**
>
> We sincerely apologize for the delay in the response. It was due to longer than expected allocation of resources and duration of several computationally-intensive experiments which were motivated by your feedback. We hope they helped to enhance our work and to address your concerns and questions. We’d like to thank you again for the review!

---

> ### Author Response · Authors · 2024-12-02
> **Summary for reviewer oGkz**
>
> Dear reviewer oGkz,
>
> As we understand your time might be limited, and given our detailed response, here we provide an executive summary of it for your convenience.
>
> ---
>
> **Q1. Why DANet is Transformer-based?** Provided a derivation from generalized attention formula to DenseAttention and highlighted minor differences across Transformer models to prove DANet is Transformer-based. Contrasted it with major differences in RNN/ SSM architectures. **(Part 1)**
>
> **W1. Related work.** Reiterated existing discussions of related work in the original manuscript. Added an in-depth analysis of Linear Transformers in relation to DenseAttention (**General Response. Part 1**) and an extended exposition of SSMs/ RNNs variants and Monarch Mixer (**Parts 1–2**) at your suggestion.
>
> **W2 and Q4. Use of self-attention in local layers/ hybrid architectures. Ablations** Explained that local layers also use DenseAttention, and thus, it’s a “pure” architecture. Drew a comparison with Gemma-2 Transformer LLM from Google. Pointed to an ablation on local attention in the paper and a new ablation. (**Part 3**)
>
> **W3. Ablations on various elements.** Explained that some ablations had failed or been impossible due to specificity of the architecture. Pointed to existing ablations/ discussions in the paper and new ablations inspired by your feedback. (**Part 3**)
>
> **W4. Performance on long contexts.** Discussed the essence of the LRA suite of benchmarks as a challenging test for long context (up to 16K length) capabilities and reiterated on remarkable DANet performance on it. Added comparisons with 25+ models on the LRA (trends hold). Presented an experiment on Pathfinder-256 (65K length) benchmark where DANet establishes a new SOTA. Conducted an ablation study by using local attention with DANet-BERT MLM and demonstrated that the model’s performance even on 16K length quickly matches the target quality which had been established on small contexts. (**Parts 4-5**)
>
> **W5. Efficiency gains of Cosine RelPE vs RoPE.** Made an ablation study and demonstrated Cosine RelPE are faster during training and inference up to 49% and 75%. (**Part 5**)
>
> **W6. Conclusion.** Presented a “Conclusion & Future Work” section. (**Part 6**)
>
> **Q2 & Q3. Differences in training and architecture between DANet and original BERT.** Explained that first 2 of 4 training stages correspond to original BERT pre-training, and the last two are for testing performance on long contexts. Fully reproduced all DANet-BERT pretraining experiments with the model accurately matching original architecture, and demonstrated that all results, relations and trends in models’ performance hold. (**Parts 6-7**)
>
> **Q5. Use of fp16.** Explained that the reasons to use it are speed gains and compatibility with older hardware. (**Part 7**)
>
> ---
>
> We express a sincere gratitude for your review which helped to enhance our work, and we are eagerly looking forward to your feedback.

---

> ### Author Response · Authors · 2024-12-03
> **Gentle reminder about discussion period end for Reviewer oGkz**
>
> Dear reviewer oGkz,
>
> We believe we have been able to carefully and comprehensively address all of your concerns and questions. With the discussion period set to close in less than 12 hours, we would greatly appreciate it if you might consider reevaluating the score of our work or sharing any additional feedback or questions, should you feel it is necessary.
>
> We thank you again for constructive feedback which led to several new experiments, ablations and other improvements.

---

### Official Review · Reviewer_HAtu · 2024-11-04

**Soundness:** 2
**Presentation:** 2
**Contribution:** 2
**Rating:** 3
**Confidence:** 4

**Summary:**

The paper proposes a DenseAttention Network (DANet), which addresses inefficiencies in the Transformer architecture, especially its high memory and computational cost - O(N^2), with respect to sequence length - N. DANet uses a new MaxNormActivation and Cosine Relative Positional Embeddings, capturing N x N interactions at O(N) space and time complexity. Experimental results demonstrate that DANet outperforms FlashAttention on long sequences on the Long Range Arena benchmark.

**Strengths:**

1) The paper proposes an interesting approach to eliminate projection matrices in attention, considering that the multiplication $W_QW_K^{\top}$ can be replaced with a single parameter, which I don't think exists in previous literature.
2) The paper also proposes using local attention in conjunction with the proposed attention function.

**Weaknesses:**

1) The paper lacks comparison against mamba in experiments. Mamba-I and Mamba-II are fast approaches for long range sequence modeling.
2) This is not the first paper which captures NXN correlations with O(N) complexity. Linear attention [1] uses linear approximations of attention. A fair comparison with this paper would be great.
3) The mathematical writing in this paper is inconsistent. Here are some instances:


Better Notation:
1. Standard operators max, var should be mentioned in times new roman using \DeclareMathOperator
2. Defined operators such as MaxNormActivation can be put in \text{MaxNormActivation}, as done in 200-204.
3. Line 240: has a typo open bracket.
4. Line 284: << should be \ll.
5. Line 246: why is fp16 and bf16 bolded?

Major readability issues:
1. Inconsistent definition of $X_i$ in line 247, 300 and 311.

If the above issues are resolved I am willing to increase my score.

[1] Katharopoulos, Angelos, et al. "Transformers are rnns: Fast autoregressive transformers with linear attention." International conference on machine learning. PMLR, 2020.

**Questions:**

See weaknesses.

---

> ### Author Response · Authors · 2024-11-25
> **Author response. Part 1**
>
> We thank the reviewer HAtu for thorough review, attention to details, and constructive feedback. We are encouraged by your recognition of the approach to eliminate the two low-rank matrices $\mathbf{W}_Q$ and $\mathbf{W}_K$ in favor of single high-rank matrix, which served as one of the reasons to name the method DenseAttention. Please let us address the concerns you have raised.
>
>
>
> > **W1. The paper lacks comparison against mamba in experiments. Mamba-I and Mamba-II are fast approaches for long range sequence modeling.**
>
> We thank you for bringing up a discussion about Mamba. We believe comparing with Mamba on the benchmarks referred to in our paper would be a delicate issue due to the following reasons:
>
> **1. Absence of LRA tests by Mamba's authors**
>
> The authors of the Mamba didn’t provide the results on the Long Range Arena (LRA) suite of benchmarks in the paper, and, it seems, they do not intend the model to be tested on the LRA. To quote the author of the paper (source: https://github.com/state-spaces/mamba/issues/282#issuecomment-2221135197), “We did not try LRA with Mamba. We don't believe that it's a good dataset, e.g. see: [1]“
>
> Despite that, we found the Mamba’s results on the LRA in another paper [2]. We compare them with DenseAttention below:
>
>
>
> | Model             | ListOps | Text   | Retrieval | Image  | Pathfinder | Path-X | Avg   |
> |-------------------|---------|--------|-----------|--------|------------|--------|-------|
> | DenseAttention    | 50.50   | 81.19 | 87.51     | 72.55  | 87.40      | 88.82  | 75.83 |
> | Mamba             | 38.02   | 82.98 | 72.14     | 69.82  | 69.26      | 67.32  | 66.59 |
>
> The metric for all tasks is accuracy, in %. Larger is better. DenseAttention significantly outperforms Mamba on average and in each benchmark individually.
>
> However, we don’t believe it might be a fair comparison because 1) the authors explicitly stated that this model is best suited for other causes, and 2) Mamba is an instance of a class of State Space Models (SSMs), whereas our model is Transformer-based.
>
> It is worth noting that, generally, SSMs are in the league of their own in comparison with Transformer-based architectures and greatly outperform them on the LRA benchmarks due to their inherent inductive bias towards capturing hierarchical and long-range dependencies and lack of such bias in Transformers, as discussed in [1, 3-4]. Nevertheless, we found that DenseAttention also outperforms one of these much stronger SSM baselines in 4 of 6 benchmarks (S4-v1 [5], results reported in the Table 1 of the paper). To the best of our knowledge, this is the first case when a pure Transformer-based model compares favorably with an SSM on the LRA which is a valuable and interesting insight.
>
> **2. Mamba's incompatibility with bidirectional sequence processing**
>
> Our second group of experiments involves pre-training BERT-like architectures with DenseAttention. We are not able to draw comparisons with Mamba, because the task for the experiments – Masked Language Modeling – requires bidirectional architecture while Mamba is a solely unidirectional, left-to-right type of model. The authors of the paper have not provided an official bidirectional implementation as of now (https://github.com/state-spaces/mamba/issues/99). Devising a bidirectional Mamba-based architecture for general sequence processing would be an important piece of work on its own merits, worth a dedicated research paper, however it’s out of scope of our current work as it is focused on improving the Transformer-based architecture.
>
> Also, we believe it’s natural to compare our models with BERT [6] specific architecture itself because we closely follow the implementation details and training process for the original model, except for replacing Transformer blocks with DANet blocks. Our key goal was to show that DenseAttention-BERT is at least on par with the original model in terms of LM quality, while being faster and more computationally efficient, and we successfully accomplished this goal.
>
> References:
>
> [1] Amos et al., "Never Train from Scratch: Fair Comparison of Long-Sequence Models Requires Data-Driven Priors." ICLR 2024
>
> [2] Alonso et al., "State Space Models as Foundation Models: A Control Theoretic Overview." arXiv preprint arXiv:2403.16899, 2024.
>
> [3] Ma et al., "Mega: Moving Average Equipped Gated Attention." ICLR 2023
>
> [4] Tran et al., "The Importance of Being Recurrent for Modeling Hierarchical Structure." EMNLP 2018
>
> [5] Gu et al., "Efficiently Modeling Long Sequences with Structured State Spaces." ICLR 2022
>
> [6] Devlin et al., "BERT: Pre-training of Deep Bidirectional Transformers for Language Understanding." NAACL 2019

---

> ### Author Response · Authors · 2024-11-25
> **Author response. Part 2**
>
> > **W2. This is not the first paper which captures NXN correlations with O(N) complexity. Linear attention [1] uses linear approximations of attention. A fair comparison with this paper would be great.**
>
> We thank you for the constructive suggestion to compare DenseAttention with the paper [1] which was one of the first to bring the concept of Linear and Linearized Transformers to light.
>
> Actually, we cited this and several other papers on linear and sub-quadratic Transformers but initially abstained from more evolved analysis. The reason is we believe that DenseAttention architecture is substantially different from Linear Transformer and its derivatives. Moreover, DenseAttention significantly outperforms it on the LRA suite of benchmarks (the comparison is presented below).
>
>
> | Model             | ListOps | Text   | Retrieval | Image  | Pathfinder | Path-X | Avg   |
> |-------------------|---------|--------|-----------|--------|------------|--------|-------|
> | Linear Transformer | 16.13   | 65.90 | 53.09     | 42.34  | 75.30      | -      | 50.55 |
> | DenseAttention    | 50.50   | 81.19 | 87.51     | 72.55  | 87.40      | 88.82  | 75.83* |
>
> \* Average for DenseAttention in this table is calculated without Path-X task for compatibility.
>
> However, motivated by your and other reviewers’ suggestions, we wrote up an extended discussion comparing Linear Transformer and its derivatives with DenseAttention Network from the view of architecture. We present it below in full and will include it in the revised version of the paper.
>
> ---
>
> Given entries $Q_i, K_j, V_j \in \mathbb{R}^{1 \times d}$ of  matrices $\mathbf{Q}, \mathbf{K}$ and $\mathbf{V}$, standard softmax attention for input i can be reformulated as
>
>
> $
> \begin{equation}
>  A_i = \frac{\sum_{j=1}^N \text{Sim}(Q_i, K_j)V_j}{\sum_{j=1}^N \text{Sim}(Q_i, K_j)} \in \mathbb{R}^{1 \times d}
> \end{equation},
> $
>
> where $\text{Sim}(Q_i, K_j)=\text{exp}(Q_i K_j^\top)$. Conceptually, linear attention class of algorithms, described in [1] and built upon in numerous subsequent works, approximates or replaces this similarity function with separable kernel $\text{Sim}(Q_i, K_j)=\mathcal{K}(Q_i, K_j)=\phi(Q_i) \phi(K_j^\top)$, where $\phi: \mathbb{R}^d \to \mathbb{R}_{+}^{r}$ maps query and key vectors to non-negative vectors with possibly different dimension r.
>
> Hence, the attention mechanism becomes:
>
>  $A_i = \frac{\sum_{j=1}^N \phi(Q_i) \phi(K_j^\top)V_j}{\sum_{j=1}^N \phi(Q_i) \phi(K_j^\top)} = \frac{\phi(Q_i) \sum_{j=1}^N  \phi(K_j^\top)V_j}{\phi(Q_i) \sum_{j=1}^N  \phi(K_j^\top)}, \quad (1)$
>
> which can be computed in linear time.
>
> The function $\phi(\cdot)$ can take various forms, such as 1 + ELU [1], ReLU [2], squared ReLU [3], Taylor [4-6] or Random Feature [7-8] approximations, and even MLPs trained to mimic softmax attention [6]. They aim to approximate softmax without its explicit calculation when being applied jointly to queries and keys, or to retain its properties, most importantly, non-negativity of resulting dot products $\phi(Q_i) \phi(K_j^\top)$.
>
>
> The latter property, together with reweighting attention scores (denominator in the formula 1) are defining for Linear Transformer algorithms. Absence of scaling by $\frac{1}{\phi(Q_i) \sum_{j=1}^n  \phi(K_j^\top)}$ leads to numerical instabilities, and the scaling factor itself is not guaranteed to be bounded without non-negative $\phi(\cdot)$.  However, both mappings $\phi(\cdot)$ (even relatively simple), and memory intensive non-MatMul operations for reweighting contribute to subpar speed and computational efficiency in comparison with ordinary and fast self-attention algorithms on all but large context sizes.
>
> We forgo both transforming $\mathbf{Q}, \mathbf{K}$ by $\phi(\cdot)$ and reweighting in DenseAttention as we believe the main factor of success of Transformer is the ability of all $N \times N$ interactions between tokens. It results in an improved computational efficiency and simpler design which can be expressed entirely by matrix multiplications:
>
> $\mathbf{A} = \mathbf{Q} \mathbf{K}^\top \mathbf{V} $
>
> [1] Katharopoulos et al., "Transformers are RNNs: Fast Autoregressive Transformers with Linear Attention." ICML 2020
>
> [2] Qin et al., "cosFormer: Rethinking Softmax in Attention." ICLR 2022
>
> [3] Hua et al., "Transformer Quality in Linear Time." ICML 2022
>
> [4] Keles et al., "On The Computational Complexity of Self-Attention." International Conference on Algorithmic Learning Theory 2023
>
> [5] Arora et al., "Simple Linear Attention Language Models Balance the Recall-Throughput Tradeoff." ICLR 2024
>
> [6] Zhang et al., "The Hedgehog & the Porcupine: Expressive Linear Attentions with Softmax Mimicry." ICLR 2024
>
> [7] Choromanski et al., "Rethinking Attention with Performers." ICLR 2021
>
> [8] Peng et al., "Random Feature Attention." ICLR 2021
>
> ---

---

> ### Author Response · Authors · 2024-11-25
> **Author response. Part 3**
>
> > **W2. Continuation**
>
> We thank you for the valuable feedback which let us enhance the exposition of our work and leave a future reader with no questions regarding the difference between Linear Transformers and DANet.
>
> We would also like to emphasize that, although it may be only subtly hinted at in the initial version of the manuscript (lines 424-425), by achieving superior results than our baseline from “Never Train from Scratch”, we automatically outperform all other Transformer-based models, which we are aware of. Now, we present these results explicitly in the following table:
>
> ---
>
> **Additional comparisons with Transformer-based models on the LRA**
>
>
> ---
>
> **Update:** We have moved the table with the additional comparisons to the **General Response** section above. Please find it attached there.
>
> ---

---

> ### Author Response · Authors · 2024-11-25
> **Author response. Part 4**
>
> > **W3**
>
> We apologize for all the typographical errors and inconsistencies and thank you for pointing them out. We really appreciate your effort and time in reviewing the manuscript, and we will fix these typos along with others, which we found during proofreading, in the revised version of the paper.

---

> ### Author Response · Authors · 2024-11-30
> **Follow-Up for Reviewer HAtu**
>
> Dear reviewer HAtu,
>
> We thank you again for the thorough review and constructive feedback which helped to improve our work.
>
> We would like to follow up and gently ask if we were able to address your concerns in the Author Response. As the discussion period progresses, we would appreciate any updates or further questions you may have. Thank you for your time in advance!

---

> ### Author Response · Authors · 2024-12-02
> **Summary for reviewer HAtu**
>
> Dear reviewer HAtu,
>
> Acknowledging that you may have time constraints, we have prepared an executive summary of our detailed response for your convenience.
>
> ---
>
> **W1. Comparisons against Mamba.** Presented comparisons with Mamba on the LRA which demonstrate DenseAttention is a clear winner. Reiterated about the comparison with another strong SSM baseline. Discussed the limitations of Mamba for bidirectional MLM and explained that our experiment design was to pre-train an architecture very closely matching BERT for fair comparison. (**Part 1**)
>
> **W2. Comparisons with Linear Transformers**. Provided an in-detail “Discussion of Linear Transformers in relation to DenseAttention”, in which we described theoretical differences between DenseAttention and the broad class of Linear Transformer models. (**Part 2**, same in **General Response. Part 1** above and in **Appendix D** in the revised manuscript)
>
> Presented comparisons with Linear Transformer and 25+ other Transformer-based architectures, including the most recent, to explicitly demonstrate that DenseAttention outperforms all of them. Restated that, to the best of our knowledge, Dense attention holds the top place on the LRA across all such architectures to date. (**Parts 2-3** and **General Response. Part 2**, also in **Appendix E.2** in the revised manuscript)
>
> **W3. Typos.** Fixed all errors and inconsistencies throughout the paper.
>
> ---
>
> We are grateful for your effort and suggestions which helped to enhance our work, and we are very keen to receive your feedback.

---

> > ### Comment · Reviewer_HAtu · 2024-12-02
> >
> > I highly recommend you conduct fair comarisons with Mamba in your future revisions. I decided to maintain my score.

---

> > > ### Author Response · Authors · 2024-12-02
> > > **On misunderstanding of Mamba fairness for Reviewer HAtu**
> > >
> > > It seems there might be a misunderstanding of "fairness" shade of meaning in our response. Our comparison is indeed fair. However, the authors of the Mamba paper themselves seem to be opposed to testing and comparing the Mamba model on the LRA as discussed above. In that sense, it might be not fair and, maybe, even unethical to them to compare their model with. That was what we meant.
> > >
> > > Since this misunderstanding appears to be resolved now by our explanation, and since we have comprehensively addressed all other your concerns, we kindly ask you to consider reevaluating our work positively or providing additional feedback, if needed.

---

### Official Review · Reviewer_ruBJ · 2024-11-04

**Soundness:** 2
**Presentation:** 3
**Contribution:** 2
**Rating:** 5
**Confidence:** 3

**Summary:**

In this paper, the authors propose a new architecture, DenseAttention Network, which could potentially replace Transformer. The motivation of this new design is to alleviate the quadratic time complexity in sequence length as well as the memory-bound operations in the vanilla Transformer (e.g. softmax and layer normalization). Specifically, they propose to linearize the original multi-head attention layer with naive matrix multiplications. To stabilize the forward pass, the inputs of each layer are scaled to have the same $\ell_\infty$ norm. The authors further propose a replacement for the rotary embedding and sliding window that is compatible with their approach.

**Strengths**
1. Given the popularity of Transformer models, the topic of their efficiency becomes more and more important. The proposed solution is also well-motivated.
2. The paper is well-written and easy to follow.

**Weaknesses**
1. The major flaw of this paper is the thin experiments.
2. The paper lacks several important previous papers.

In summary, this paper proposes a potential solution to accelerate Transformer models. However, the experiments are not convincing enough. Therefore, I would recommend a clear rejection unless there is further evidence.

**Strengths:**

1. Given the popularity of Transformer models, the topic of their efficiency becomes more and more important. The proposed solution is also well-motivated.
2. The paper is well-written and easy to follow.

**Weaknesses:**

1. The major flaw of this paper is the thin experiments. The Transformer model is known to perform well on a wide range of tasks. In addition, it also demonstrates a promising scaling effect. On the other hand, this paper only contains limited experiments: (1) the testbeds are limited. Currently, the only benchmark is Long Range Arena (for causal LMs); (2) the baselines are limited. There is only one Transformer model that serves as the baseline without specifying how the model is trained; (3) the scaling effect is not studied. The authors do not analyze how the parameter number affects the results. It is unclear if the method could be scaled to larger-scale applications.
2. The paper lacks several important previous papers. In fact, linearizing attention has been heavily studied before [1, 2, 3]. This paper has no comparisons or discussions.

[1] Random Feature Attention, ICLR 2021 \
[2] Transformers are RNNs: Fast Autoregressive Transformers with Linear Attention, ICML 2020 \
[3] Transformer Dissection: A Unified Understanding of Transformer's Attention via the Lens of Kernel, EMNLP 2019

**Questions:**

See weaknesses.

---

> ### Author Response · Authors · 2024-11-25
> **Author response. Part 1**
>
> We thank the reviewer ruBJ for insightful comments  and valuable feedback. We are grateful for your recognition of the motivation and the presentation of the paper. In the following, we hope to address your concerns with detailed responses:
>
> > **W1.1 The testbeds are limited. Currently, the only benchmark is Long Range Arena (for causal LMs)**.
>
> We thank you for bringing up this issue. We think there might be some confusion about the scope of the experiments and we would like to clarify it with an extended discussion:
>
> **1.** The Long Range Arena [3] (section 4.1) is not the only benchmark used in the paper. We also conduct extensive experiments and benchmarks on Masked Language Modeling (MLM) in section 4.2 with BERT-styled DANet models as compared to standard Transformer-based BERT.
>
> **2.** The Long Range Arena is, in fact, not a single benchmark but a suite of *6 challenging and diverse tasks* designed to test modeling capabilities across different domains. Below is a brief description of each task.
>
> ---
> **ListOps.**[4] This is a purely logical synthetic task which is dedicated to modeling evaluation results of long hierarchically structured sequences. Each sequence has length up to 2000 symbols and consists of whole numbers from 0 to 9, mathematical operators, such as MAX, MIN, MEDIAN and SUM_MOD, and parentheses.
>
> **Text Classification (IMDB)** [5]. This task tests Natural Language Understanding (NLU) abilities of models by letting them classify the sentiment of movie reviews in the IMDB dataset. To make the task more challenging, the texts of the reviews are split into tokens not on a word level, but on a character (or byte) level. This leads to much longer sequences of 4K max length.
>
> **Document Retrieval (AAN)** [6]. This task tests the abilities of producing encoded representations of the textual information and further matching/ retrieving them. Namely, given a pair of the documents from ACL Anthology Network (AAN; Radev et al., 2013) dataset, a model should independently process them and, based on their final embeddings, classify if the two documents have a citation link. As in the IMDB tasks, individual input texts are tokenized on a character (byte) level with max sequence length 4K.
>
> **Image Classification (CIFAR-10)** [7]. This is an image classification task with 10 classes on a classical CIFAR-10 benchmark with one specific condition: images should be ingested into models as 1-d sequences, thus setting the input length to 1024 tokens (pixels) and making the task more challenging.
>
> **Pathfinder** [8]. This is a binary classification task of 32x32 pixels grayscale images with corresponding sequence length 1024 tokens, which, formally, makes it similar to CIFAR-10 task. However, it’s different on a conceptual level, as the task measures a model’s ability to discern spatial dependencies. Given a multitude of intertwined, dashed line paths, a model should correctly determine if two rounded dots are connected by a dashed line.
>
> **Pathfinder-X (Pathfinder-128)**. It’s a version of Pathfinder task with 16K (128x128) pixels images which makes it significantly more challenging. At the time of publication of the original LRA paper [3], none of the tested models managed to achieve a score above chance on this benchmark.
>
> Therefore, the Long Range Arena arguably represents a wide range of tasks, spanning from logic and reasoning to language modeling and image classification. To perform well on all of the 6 benchmarks, a model’s architecture should be powerful and versatile enough to generalize to different modalities.
>
> ---
>
> We thank you for the opportunity to clarify and enhance the exposition of the experiments section. We will include the description of the LRA benchmarks in the appendix of the new version of  manuscript.
>
> ---
> **3.** We would also like to kindly note that the amount of the experimentation in our work seems to be adequate in comparison with many other papers introducing novel architectural modifications of Transformers and published in leading ML venues.
>
> For example, the ICLR-published paper “Random Feature Attention” [2], which you have referenced in the review, contains experiments with Language Modeling (causal and machine translation) and Long Range Arena, albeit they tested their model only on 3 out of 6 its benchmarks.
>
> Another widely known paper “Transformers are RNNs” [1], which you have also referenced, contains experiments on autoregressive image generation (MNIST and CIFAR-10), speech recognition, and artificial copying task. However, it doesn’t even provide experiments or results for language modeling task. Also, there’s no Long Range Arena results in the paper, although it can be attributed to the fact that this suite of benchmarks was not presented at that time.
>
> ---

---

> ### Author Response · Authors · 2024-11-25
> **Author response. Part 2**
>
> > **W1.1 (Continuation)**
>
> **4.** Moreover, motivated by the suggestions, we decided to conduct an additional **experiment on the Pathfinder-256 task**. This is an extremely challenging version of the Pathfinder task with sequence length 65k which is on par with input context size of recent generations of proprietary Large Language Models. We present the results below:
>
> ---
>
> | Algorithm                                | Accuracy on the validation set, % |
> |-----------------------------------------|------------------------------------|
> | FlashAttention [9]                      | 63.1                               |
> | S4 [11]                                 | 67.8                               |
> | DenseAttention                          | _72.6_                               |
> | DenseAttention after additional 550 epochs | **77.1**                            |
>
>
> DenseAttention model outperforms existing results from the literature of standard Transformer augmented with FlashAttention [9] and S4-v2 model [12] as reported in [11] The result holds both when the training procedure is carried out for 200 training epochs as in [9] and then it’s prolonged for 550 additional epochs.
>
> This experiment lets us make several observations:
>
> * DenseAttention Network architecture performs well even on very long input sequences which is promising given current trend of increasing context size in modern Large Language and Multimodal Models
> * DenseAttention shows favorable scaling properties with respect to the amount of training iterations, even with the fixed dataset size. * The validation accuracy for the task kept improving throughout the whole training and would likely have continued if the experiment had not been stopped.
> * Truly linear scaling in sequence length is crucial for improvements in quality for large contexts. It took approximately 3 days on 4 H100 GPUs to train our model for 750 epochs in linear mode, while the projected runtime of quadratic (FlashAttention-2 [10]) and log-linear (S4) algorithms in the same setting would be at best 3 and 0.5 months, respectively, which renders them impractical for prolonged training.
> ---
>
> **5)** We acknowledge that there are other modalities and specialized architectures that would benefit from long-context efficiency improvements if the DenseAttention is ported or applied to them, such as ViT and SAM for Computer Vision tasks and LLAMA for language modeling. We hope to address them in future work.
>
> [1] Katharopoulos et al., "Transformers are RNNs: Fast Autoregressive Transformers with Linear Attention." ICML 2020
>
> [2] Peng et al., "Random Feature Attention." ICLR 2021
>
> [3] Tay et al., "Long Range Arena: A Benchmark for Efficient Transformers." ICLR 2021
>
> [4] Nangia and Bowman, "ListOps: A Diagnostic Dataset for Latent Tree Learning." NAACL 2018
>
> [5] Maas et al., "Learning Word Vectors for Sentiment Analysis." ACL 2011
>
> [6] Radev et al., "The ACL Anthology Network Corpus." Language Resources and Evaluation, 2013
>
> [7] Krizhevsky, "Learning Multiple Layers of Features from Tiny Images." Technical Report, 2009
>
> [8] Kim et al., "Disentangling Neural Mechanisms for Perceptual Grouping." ICLR 2020
>
> [9] Dao et al., "FlashAttention: Fast and Memory-Efficient Exact Attention with IO-Awareness." NeurIPS 2022
>
> [10] Dao, T., "FlashAttention-2: Faster Attention with Better Parallelism and Work Partitioning." ICLR 2024
>
> [11] Amos et al., "Never Train from Scratch: Fair Comparison of Long-Sequence Models Requires Data-Driven Priors." ICLR 2024
>
> [12] Gu et al., "Efficiently Modeling Long Sequences with Structured State Spaces." ICLR 2022
>
> ---
>
> > **W1.2 The baselines are limited. There is only one Transformer model that serves as the baseline without specifying how the model is trained.**
>
> We thank you for pointing out the perceived lack of baselines. Indeed, Table 1 in the paper lists only two baselines. However, these baselines are exceptionally strong. Actually, DANet outperforms *all transformer-based models and architectures that we are aware of*, including recent ones, such as [10-11] on the Long Range Arena suite of benchmarks.
>
> **Additional baselines**
>
> We acknowledged the strength of our main baseline from (Never Train from Scratch, ICLR 2024)  in the lines 424-425, quote: “Interestingly, even without pre-training, but with RoPE, they reached a SOTA score on the benchmark among all Transformer-based architectures with a large margin”. In hope that it would be clear from the text, and due to space limitations, we omitted explicit comparisons with other 25+ models, including 11 models tested in the original LRA paper (reference), in the Table 1. We present the full comparisons below in a separate table.

---

> ### Author Response · Authors · 2024-11-25
> **Author response. Part 3. Additional baselines**
>
> > **W1.2 (Continuation)**
>  ---
>
> **Update:** We have moved the table with the additional baselines to the **General Response** section above. Please find it attached there.
>
> ---

---

> ### Author Response · Authors · 2024-11-25
> **Author response. Part 4**
>
> > **W1.2 (Ending)**
>
> ---
>
> **Discussion of the main baselines**
>
> “Transformers + Rotary” result from recent ICLR-published paper “Never Train from Scratch” which serves as our primary baseline, in its turn, outperforms all previous Transformer-based models by a large margin. The details how the model is trained are comprehensively described both in the paper (https://openreview.net/pdf?id=PdaPky8MUn) and in the code (https://github.com/IdoAmos/not-from-scratch).
>
> We would like to emphasize that DenseAttention outperforms, among others, Linear Attention [1] and Random Feature Attention [2] architectures, which you suggested to compare with in W2.
>
> As you have correctly noticed, our second baseline for the LRA – S4 model [3] – is not a Transformer based architecture. It belongs to a class of State Space Models (SSMs) which almost uniformly greatly outperform Transformer-based models on this suite of benchmarks due to their inherent inductive bias towards capturing hierarchical and long-range dependencies and lack of such bias in Transformers [4-7]. This makes our result superior to an instance of SSM in several benchmarks a valuable and interesting insight.
>
> ---
>
> We are grateful to you for the chance to clarify and improve the exposition of our work. We will include the table with the extended LRA results in the appendix of the next version of the manuscript.
>
> ---
>
> **BERT experiments**
>
> Regarding the experiments with BERT-like architecture, we believe it’s natural to compare our models with BERT itself as we closely follow the implementation details and training process for the original model, except for replacing Transformer blocks with DANet blocks. Our key objective is to show that DenseAttention-BERT is at least on par with the original model in terms of LM quality, while being faster and more computationally efficient. Please note that we demonstrated the latter property by comparing our plain-PyTorch implementation of DANet both with popular PyTorch implementation by HuggingFace and with low-level specialized CUDA implementation FlashAttention-2 [7], widely regarded as the fastest attention computation algorithm available and universally used in practice (https://pytorch.org/docs/stable/generated/torch.nn.functional.scaled_dot_product_attention.html).
>
> ----
>
> [1] Katharopoulos et al., "Transformers are RNNs: Fast Autoregressive Transformers with Linear Attention." ICML 2020
>
> [2] Peng et al., "Random Feature Attention." ICLR 2021
>
> [3] Gu et al., "Efficiently Modeling Long Sequences with Structured State Spaces." ICLR 2022
>
> [4] Amos et al., "Never Train from Scratch: Fair Comparison of Long-Sequence Models Requires Data-Driven Priors." ICLR 2024
>
> [5] Ma et al., "Mega: Moving Average Equipped Gated Attention." ICLR 2023
>
> [6] Tran et al., "The Importance of Being Recurrent for Modeling Hierarchical Structure." EMNLP 2018
>
> [7] Dao, T., "FlashAttention-2: Faster Attention with Better Parallelism and Work Partitioning." ICLR 2024
>
> ---
>
> > **W1.3 The scaling effect is not studied. The authors do not analyze how the parameter number affects the results. It is unclear if the method could be scaled to larger-scale applications.**
>
> We thank you for the constructive feedback. Inspired by it, we conducted an additional experiment on scaling, which results are presented below.
>
> | Model              | Parameters | Configuration   | Perplexity | MLM accuracy, % |
> |--------------------|------------|------------------|------------|-----------------|
> | DANet-BERT-small   | 31M        | L=6, D=512      | 15.60      | 49.51           |
> | DANet-BERT-base    | 110M       | L=16, D=768     | 7.55       | 60.01           |
> | DANet-BERT-large   | 336M       | L=32, D=1024    | 5.47       | 64.92           |
>
> The table depicts three single-head DenseAttention Network models of different sizes pre-trained on Wiki+BookCorpus dataset with MLM objective for 100B tokens. MLM perplexity and accuracy are reported for out-of-sample data from C4 dataset [1]. L and D denote number of layers and hidden dimension of FFN input, respectively.
>
> We observe that DenseAttention architecture exhibits favorable scaling properties similar to vanilla Transformer, as modeling quality grows with the parameters count. Also, as we show in the Pathfinder-256 experiment, the quality consistently increases with the number of training iterations, which opens a promising second axis for scaling.
>
> However, we acknowledge that the architecture hasn't been tested on billion+ parameters range models, and we hope to explore the applications to LLMs in future work.
>
> ---
> [1] Raffel et al., "Exploring the Limits of Transfer Learning with a Unified Text-to-Text Transformer." JMLR 2020

---

> ### Author Response · Authors · 2024-11-25
> **Author response. Part 5**
>
> > **W2. The paper lacks several important previous papers. In fact, linearizing attention has been heavily studied before [1, 2, 3]. This paper has no comparisons or discussions.**
>
> We thank you for pointing out limited discussions and comparisons with other papers related to linearized attention. In fact, we cited several papers on linear and sub-quadratic Transformers, including one of the first and most influential works on linear attention [1] in the Introduction (lines 52-53), and LocalAttention for DenseAttention (lines 363-364; 375 where we discussed similarities with our local attention paradigm) sections.
>
> However, we initially omitted the extended discussion of [1] and subsequent work as we believe DenseAttention architecture is conceptually different from Linear Transformer and its derivatives. The most similar research to ours is SimA [4], which we discuss in the Introduction.
>
> As we commented in response to W1.2, DenseAttention compares favorably with all Transformer-based architectures we are aware of and which have been tested on the LRA, including linear and subquadratic algorithms. In particular, the model outperforms both Linear Transformer [1] and Random Feature Attention [2] by a large margin. We copy select rows from the table for convenience:
>
> | Model               | ListOps | Text   | Retrieval | Image  | Pathfinder | Avg    |
> |---------------------|---------|--------|-----------|--------|------------|--------|
> | Linear Trans.    | 16.13   | 65.90  | 53.09     | 42.34  | 75.30      | 50.55  |
> | RFA             | 36.80   | 66.00  | 56.10     | -      | -          | -      |
> | DenseAttention      | 50.50   | 81.19  | 87.51     | 72.55  | 87.40      | 75.83  |
>
> [1] Katharopoulos et al., "Transformers are RNNs: Fast Autoregressive Transformers with Linear Attention." ICML 2020
>
> [2] Peng et al., "Random Feature Attention." ICLR 2021
>
> [3] Tsai et al., "Transformer Dissection: A Unified Understanding of Transformer's Attention via the Lens of Kernel." EMNLP 2019
>
> [4] Koohpayegani and Pirsiavash, "SimA: Simple Softmax-free Attention for Vision Transformers." WACV 2024
>
> ---
>
> Moreover, based on the reviews, we understand now that explicit analysis of [1] would greatly improve our work and sincerely thank you for the suggestion. We will incorporate the following discussion in the paper.
>
> ---
>
> **Discussion of Linear Transformers in relation to DenseAttention**
>
> Given entries $Q_i, K_j, V_j \in \mathbb{R}^{1 \times d}$ of  matrices $\mathbf{Q}, \mathbf{K}$ and $\mathbf{V}$, standard softmax attention for input i can be reformulated as
>
>
> $
> \begin{equation}
>  A_i = \frac{\sum_{j=1}^N \text{Sim}(Q_i, K_j)V_j}{\sum_{j=1}^N \text{Sim}(Q_i, K_j)} \in \mathbb{R}^{1 \times d}
> \end{equation},
> $
>
> where $\text{Sim}(Q_i, K_j)=\text{exp}(Q_i K_j^\top)$. Conceptually, linear attention class of algorithms, described in [1] and built upon in numerous subsequent works, approximates or replaces this similarity function with separable kernel $\text{Sim}(Q_i, K_j)=\mathcal{K}(Q_i, K_j)=\phi(Q_i) \phi(K_j^\top)$, where $\phi: \mathbb{R}^d \to \mathbb{R}_{+}^{r}$ maps query and key vectors to non-negative vectors with possibly different dimension r.
>
> Hence, the attention mechanism becomes:
>
>  $A_i = \frac{\sum_{j=1}^N \phi(Q_i) \phi(K_j^\top)V_j}{\sum_{j=1}^N \phi(Q_i) \phi(K_j^\top)} = \frac{\phi(Q_i) \sum_{j=1}^N  \phi(K_j^\top)V_j}{\phi(Q_i) \sum_{j=1}^N  \phi(K_j^\top)}, \quad (1)$
>
> which can be computed in linear time.
>
> The function $\phi(\cdot)$ can take various forms, such as 1 + ELU [1], ReLU [2], squared ReLU [3], Taylor [4-6] or Random Feature [7-8] expansions, and even MLPs trained to mimic softmax attention [6]. They aim to approximate softmax without its explicit calculation when being applied jointly to queries and keys, or to retain its properties, most importantly, non-negativity of resulting dot products $\phi(Q_i) \phi(K_j^\top)$.
>
>
> The latter property, together with reweighting attention scores (denominator in the formula 1) are defining for Linear Transformer algorithms. Absence of scaling by $\frac{1}{\phi(Q_i) \sum_{j=1}^n  \phi(K_j^\top)}$ leads to numerical instabilities, and the scaling factor itself is not guaranteed to be bounded without non-negative $\phi(\cdot)$.  However, both mappings $\phi(\cdot)$ (even relatively simple), and memory intensive non-MatMul operations for reweighting contribute to subpar speed and computational efficiency in comparison with ordinary and fast self-attention algorithms on all but large context sizes.
>
> We forgo both transforming $\mathbf{Q}, \mathbf{K}$ by $\phi(\cdot)$ and reweighting in DenseAttention as we believe the main factor of success of Transformer is the ability of all $N \times N$ interactions between tokens. It results in an improved computational efficiency and simpler design which can be expressed entirely by matrix multiplications:
>
> $\mathbf{A} = \mathbf{Q} \mathbf{K}^\top \mathbf{V} $

---

> ### Author Response · Authors · 2024-11-25
> **Remaining references**
>
> [1] Katharopoulos et al., "Transformers are RNNs: Fast Autoregressive Transformers with Linear Attention." ICML 2020
>
> [2] Qin et al., "cosFormer: Rethinking Softmax in Attention." ICLR 2022
>
> [3] Hua et al., "Transformer Quality in Linear Time." ICML 2022
>
> [4] Keles et al., "On The Computational Complexity of Self-Attention." International Conference on Algorithmic Learning Theory 2023
>
> [5] Arora et al., "Simple Linear Attention Language Models Balance the Recall-Throughput Tradeoff." ICLR 2024
>
> [6] Zhang et al., "The Hedgehog & the Porcupine: Expressive Linear Attentions with Softmax Mimicry." ICLR 2024
>
> [7] Choromanski et al., "Rethinking Attention with Performers." ICLR 2021
>
> [8] Peng et al., "Random Feature Attention." ICLR 2021

---

> ### Author Response · Authors · 2024-11-30
> **Follow-Up for Reviewer ruBJ**
>
> Dear reviewer ruBJ,
>
> We thank you again for your insightful comments and valuable feedback, which prompted us to conduct new experiments and add extended discussions for several topics.
>
> We would like to follow up and gently ask if we were able to address your concerns in the Author Response. As the discussion period progresses, we would appreciate any updates or further questions you may have. Thank you for your time in advance!

---

> > ### Author Response · Authors · 2024-12-02
> > **Summary for reviewer ruBJ**
> >
> > Dear reviewer ruBJ,
> >
> > As we acknowledge that your time may be constrained, and given the comprehensive scope of our response, here we present its summary for your convenience.
> >
> > **W1.1. Limited testbeds.** Communicated that the LRA is not the only testbed; there are also extensive experiments with standalone language modeling in the paper (**Section 4.2**). Presented an extensive discussion of diverse and comprehensive scope for all 6 of the LRA benchmarks which signifies the broad and versatile nature of the chosen testbeds (**Part 1**, also **Appendix E.1** in the revised manuscript). Drew a comparison with the experimentation scope of related published papers highlighting the same or greater breadth in our work. Conducted a new experiment on extremely challenging Pathfinder-256 task (65K context size) and set a new SOTA (**Part 2**, also **Appendix H.1**). Discussed directions for future work.
> >
> > **W1.2. Limited baselines.** Reiterated on exceptional strength of our augmented Transformer baseline on the LRA, which in turn outperforms all Transformer-based models. Referred to specifications on how it was trained. Presented explicit comparisons with 25+ Transformer-based architectures to validate that DenseAttention outperforms all of them, including the models mentioned in W2 and the most recent ones (**Parts 2-4** and **General Response. Part 2** above, also available in **Appendix E.2**). Highlighted the significance of DenseAttention also outperforming an instance of SSM class much better suited for the LRA. Explained the LM experiment design to pre-train an architecture very closely matching BERT for fair comparison in modeling performance and speed. Reported that DANet compares favorably in both. (**Part 4**)
> >
> > **W1.3. Scaling effects**
> >
> > Presented a scaling effects study w.r.t. the model size which indicates DenseAttention architecture exhibits favorable scaling properties similar to Transformer (**Part 4**, also **Appendix H.3**). Expressed promising scaling properties w.r.t. the amount of training as shown in the new Pathfinder-256 experiment.
> >
> > **W2. Discussion of Linear Transformers**
> >
> >  Reiterated existing discussions of related work in the original manuscript. Provided an in-detail “Discussion of Linear Transformers in relation to DenseAttention”, in which we described theoretical differences between DenseAttention and the broad class of Linear Transformer models (**Part 5**, same in **General Response. Part 1** and in **Appendix D** in the revised manuscript).
> >
> >
> > We thank you for your time and insightful suggestions which helped to refine our work, and we eagerly await your feedback.

---

> > > ### Comment · Reviewer_ruBJ · 2024-12-03
> > >
> > > Dear authors. Thank you for the response and the summary.
> > >
> > > > W1.1. Limited testbeds.
> > >
> > > I was aware that LRA contains multiple datasets. I expected the author to obtain more results beyond LRA. In particular, the paper does not contain the standard LM eval tasks like MMLU, GSM8K, TriviaQA, ARC, Hellaswag, MATH, etc.
> > >
> > > > W1.2. Limited baselines.
> > >
> > > Thanks for providing more baselines. The major issue in the updated table is the authors should carefully select related baselines (similar parameter counts and training data) for fair comparisons. It is not a contest to have more rows.
> > >
> > > > W1.3. Scaling effects
> > >
> > > Thanks for showing the scaling effect. However, the scaling is from an ultra-small scale (31M) to a small scale (336M). More than five years ago, the smallest GPT2 already had 100M+ parameters. In the initial review, I was concerned that "It is unclear if the method could be scaled to larger-scale applications".
> > >
> > > > W2. Discussion of Linear Transformers
> > >
> > > Thanks for adding the discussion. It is now clearer where the paper should be positioned. But using an identity mapping as $\phi$ and dropping the denominator does not seem to largely contribute to the established knowledge unless the authors could justify that previous papers were doing wrong (and therefore could explain why their performance is much lower than DenseAttention).
> > >
> > > In summary, the authors only partially addressed the concerns. In recognition of this and the authors' diligence, I have raised my score to 5.

---

> > > > ### Author Response · Authors · 2024-12-03
> > > > **Gratitude and further explanations for Reviewer ruBJ**
> > > >
> > > > Dear reviewer ruBJ,
> > > >
> > > > We sincerely thank you very much for your response! We are really delighted and encouraged to have partially addressed your concerns.
> > > >
> > > > We hope to alleviate the remaining concerns at least to some extent with the following discussion.
> > > >
> > > > ---
> > > >
> > > > > **W1.2.** The major issue in the updated table is the authors should carefully select related baselines (similar parameter counts and training data)
> > > >
> > > > All the models which are tested on the LRA should be trained from scratch on the datasets pre-supplied by the authors of this suite of benchmarks.  No additional data or pre-training is allowed as per guidelines for the LRA set by authors in the paper. They also set the requirement for the participating models to have an approximately fixed number of parameters for each task. All of the papers we compare with have explicitly stated they adhere to these requirements and have undergone peer review.
> > > >
> > > > ---
> > > >
> > > > > **W1.1.** “The paper does not contain the standard LM eval tasks like MMLU, GSM8K, TriviaQA, ARC, Hellaswag, MATH, etc.”
> > > >
> > > > > **W1.3.** “However, the scaling is from an ultra-small scale (31M) to a small scale (336M).” and “It is unclear if the method could be scaled to larger-scale applications"
> > > >
> > > > Unfortunately, our computational resources are limited which prevents us from being able to pre-train multi-billion parameter models for adequate amount of time/ iterations, both for scaling studies and for evaluating on the suggested tasks. In particular, for the majority of suggested tasks, multi-billion parameter models trained on hundreds of billions or trillions of tokens are required to score above chance.
> > > >
> > > > However, existing experiments indicate that DenseAttention is a viable and performant architecture as it performs comparably and better than standard Transformers on tested scales and exhibits upward trend with the increase of parameters, just like the Transformer. We argue that, given current evidence, there are no reasons to suspect the scaling trend will halt with large models as it does not with the standard Transformer.
> > > >
> > > >
> > > > We would also like to acknowledge that many papers which introduce new Transformer-based architectures, conduct experiments on equal or smaller scale models compared to ours. These include, among others, recent ICLR published [1-2] and ICML published [3] papers.
> > > >
> > > > Finally, as we have stated in our initial response, large scale experiments with GPT and LLAMA style architectures are out of scope for current work, however, we hope to address it in future work. To articulate the scope of the paper more clearly, we have added a “Conclusion & Future Work” section. We present it in the **General Response. Part 3** above for your convenience and also in the revised manuscript (**Appendix A**).
> > > >
> > > > [1] Zhang et al., "The Hedgehog & the Porcupine: Expressive Linear Attentions with Softmax Mimicry." ICLR 2024
> > > >
> > > > [2] Zheng et al., "Efficient Attention via Control Variates." ICLR 2023
> > > >
> > > > [3] Zandieh et al., "KDEformer: Accelerating Transformers via Kernel Density Estimation."
> > > > ICML 2023

---

> > > > > ### Author Response · Authors · 2024-12-03
> > > > > **Gratitude and further explanations for Reviewer ruBJ (2)**
> > > > >
> > > > > > W2. “But using an identity mapping as $\phi$ and dropping the denominator does not seem to largely contribute to the established knowledge unless the authors could justify that previous papers were doing wrong (and therefore could explain why their performance is much lower than DenseAttention).”
> > > > >
> > > > > Complete removal of non-trivial $\phi$ and the denominator from the attention is a rewarding, but very hard thing to achieve as explained in the paper. To the best of our knowledge, we are the first to successfully do it and demonstrate excellent modeling performance. It unlocks another conceptual paradigm since attention scores are no longer constrained to be non-negative *as in all previous Transformer-based architectures* starting from the vanilla softmax self-attention. We argue that this increased expressivity and versatility contributes positively to the performance.
> > > > >
> > > > > We emphasize that framing the absence of any transformations applied to $\mathbf{Q}$, $\mathbf{K}$, or their dot-products as an identity mapping $\phi(x)=x$ would be conceptually wrong, because attention scores are required to be non-negative both in the Linear and general Kernelized (vanilla self-attention belongs here) attention classes [1-3]. They also need to be normalized by some weights, and the absence of reweighting the attention scores by their row-wise sums further sets DenseAttention apart from all other algorithms.
> > > > >
> > > > > Both elements are memory-intensive operations and exactly justify and explain relative computational inefficiency of attention mechanisms w.r.t DenseAttention.
> > > > >
> > > > > To conclude, exactly the omission of these two elements brings both modeling quality and speed/ computational efficiency gains. However, designing such an architecture to be numerically stable and to have well-behaved activations is very hard. We believe that accomplishing it and proving the architecture performs well in different settings is a major and valuable finding.
> > > > >
> > > > > [1] Katharopoulos et al., "Transformers are RNNs: Fast Autoregressive Transformers with Linear Attention." ICML 2020
> > > > >
> > > > > [2] Choromanski et al., "Rethinking Attention with Performers." ICLR 2021
> > > > >
> > > > > [3] Tsai et al., "Transformer Dissection: A Unified Understanding of Transformer's Attention via the Lens of Kernel." EMNLP 2019
> > > > >
> > > > > ---
> > > > >
> > > > > Again, we express our gratitude for your response and positive evaluation of our efforts in the rebuttal. We also hope to have been able to alleviate your remaining concerns with new explanations. If this is the case, we kindly ask you to consider further updating your score or sharing any additional feedback or questions, should you feel it is necessary.

---

### Author Response · Authors · 2024-11-27
**General Response. Part 1. Discussion of Linear Transformers in relation to DenseAttention**

Dear Reviewers,

We sincerely thank you for the precious time, insightful feedback and constructive suggestions which helped to clarify and enhance the exposition of our work.

Here we would like to address several key commonly shared comments.

---

**Discussion of Linear Transformers in relation to DenseAttention**

The majority of reviewers suggested to put DenseAttention into perspective with Linear Transformer class of algorithms. Here’s the extended discussion comparing Linear Transformer and its derivatives with DenseAttention Network from the architectural standpoint. We present it below in full and will include it in the revised version of the paper.

Given entries $Q_i, K_j, V_j \in \mathbb{R}^{1 \times d}$ of  matrices $\mathbf{Q}, \mathbf{K}$ and $\mathbf{V}$, standard softmax attention for input i can be reformulated as


$
\begin{equation}
 A_i = \frac{\sum_{j=1}^N \text{Sim}(Q_i, K_j)V_j}{\sum_{j=1}^N \text{Sim}(Q_i, K_j)} \in \mathbb{R}^{1 \times d}
\end{equation},
$

where $\text{Sim}(Q_i, K_j)=\text{exp}(Q_i K_j^\top)$. Conceptually, linear attention class of algorithms, described in [1] and built upon in numerous subsequent works, approximates or replaces this similarity function with separable kernel $\text{Sim}(Q_i, K_j)=\mathcal{K}(Q_i, K_j)=\phi(Q_i) \phi(K_j^\top)$, where $\phi: \mathbb{R}^d \to \mathbb{R}_{+}^{r}$ maps query and key vectors to non-negative vectors with possibly different dimension r.

Hence, the attention mechanism becomes:

 $A_i = \frac{\sum_{j=1}^N \phi(Q_i) \phi(K_j^\top)V_j}{\sum_{j=1}^N \phi(Q_i) \phi(K_j^\top)} = \frac{\phi(Q_i) \sum_{j=1}^N  \phi(K_j^\top)V_j}{\phi(Q_i) \sum_{j=1}^N  \phi(K_j^\top)}, \quad (1)$

which can be computed in linear time.

The function $\phi(\cdot)$ can take various forms, such as 1 + ELU [1], ReLU [2], squared ReLU [3], Taylor [4-6] or Random Feature [7-8] expansions, and even MLPs trained to mimic softmax attention [6]. They aim to approximate softmax without its explicit calculation when being applied jointly to queries and keys, or to retain its properties, most importantly, non-negativity of resulting dot products $\phi(Q_i) \phi(K_j^\top)$.


The latter property, together with reweighting attention scores (denominator in the formula 1) are defining for Linear Transformer algorithms. Absence of scaling by $\frac{1}{\phi(Q_i) \sum_{j=1}^n  \phi(K_j^\top)}$ leads to numerical instabilities, and the scaling factor itself is not guaranteed to be bounded without non-negative $\phi(\cdot)$.  However, both mappings $\phi(\cdot)$  and memory intensive non-MatMul operations for reweighting contribute to subpar speed and computational efficiency in comparison with ordinary and fast self-attention algorithms on all but large context sizes.

We forgo both transforming $\mathbf{Q}, \mathbf{K}$ by $\phi(\cdot)$ and reweighting in DenseAttention as we believe the main factor of success of Transformer is the ability of all $N \times N$ interactions between tokens. It results in an improved computational efficiency and simpler design which can be expressed entirely by matrix multiplications:

$\mathbf{A} = \mathbf{Q} \mathbf{K}^\top \mathbf{V} $

---

**References**

[1] Katharopoulos et al., "Transformers are RNNs: Fast Autoregressive Transformers with Linear Attention." ICML 2020

[2] Qin et al., "cosFormer: Rethinking Softmax in Attention." ICLR 2022

[3] Hua et al., "Transformer Quality in Linear Time." ICML 2022

[4] Keles et al., "On The Computational Complexity of Self-Attention." International Conference on Algorithmic Learning Theory 2023

[5] Arora et al., "Simple Linear Attention Language Models Balance the Recall-Throughput Tradeoff." ICLR 2024

[6] Zhang et al., "The Hedgehog & the Porcupine: Expressive Linear Attentions with Softmax Mimicry." ICLR 2024

[7] Choromanski et al., "Rethinking Attention with Performers." ICLR 2021

[8] Peng et al., "Random Feature Attention." ICLR 2021

---

> ### Author Response · Authors · 2024-11-27
> **General Response. Part 2. Additional comparisons on the LRA (1)**
>
> **Additional comparisons of DenseAttention on the LRA**
>
> In our paper, we have stated that our baseline from [11] for the LRA suite of benchmarks surpasses all previous Transformer-based architectures we are aware of. Since DenseAttention outperforms this baseline and due to space limitations, we initially omitted explicit comparisons with these models. However, the reviewers’ comments indicated that it’s more advisable to present them.
>
> We post the comparison table here in full and will include it in the Appendix of the paper.
>
>
> | Model                 | ListOps    | Text      | Retrieval   | Image      | Pathfinder  | Path-X     | Avg        |
> |-----------------------|------------|-----------|-------------|------------|-------------|------------|------------|
> | Transformer [1]          | 36.37      | 64.27     | 57.46       | 42.44      | 71.40       | 61.40 [12]     | 54.39      |
> | Local Attention [1]      | 15.82      | 52.98     | 53.39       | 41.46      | 66.63       | -          | 46.06      |
> | Sparse Trans. [1]       | 17.07      | 63.58     | 59.59       | 44.24      | 71.71       | -          | 51.24      |
> | Longformer [1]          | 35.63      | 62.85     | 56.89       | 42.22      | 69.71       | -          | 53.46      |
> | Linformer [1]             | 35.70      | 53.94     | 52.27       | 38.56      | 76.34       | -          | 51.36      |
> | Reformer [1]             | 37.27      | 56.10     | 53.40       | 38.07      | 68.50       | -          | 50.67      |
> | Sinkhorn Trans. [1]      | 33.67      | 61.20     | 53.83       | 41.23      | 67.45       | -          | 51.29      |
> | Synthesizer [1]          | 36.99      | 61.68     | 54.67       | 41.61      | 69.45       | -          | 52.88      |
> | BigBird [1]              | 36.05      | 64.02     | 59.29       | 40.83      | 74.87       | -          | 55.01      |
> | Linear Trans. [1]         | 16.13      | 65.90     | 53.09       | 42.34      | 75.30       | -          | 50.55      |
> | Performer [1]            | 18.01      | 65.40     | 53.82       | 42.77      | 77.05       | -          | 51.41      |
> | RFA [2]                  | 36.80      | 66.00     | 56.10       | -          | -           | -          | -          |
> | Luna-256 [3]             | 37.98      | 65.78     | 79.56       | 47.86      | _78.55_     | -          | 61.95      |
> | Nyströmformer [4]        | 37.15      | 65.52     | 79.56       | 41.58      | 70.94       | -          | 58.95      |
> | Kernelized Attention [5] | 38.78      | 60.22     | 81.77       | 41.29      | 70.73       | -          | 58.56      |
> | Informer [5]             | 32.53      | 62.64     | 77.57       | 38.10      | 57.83       | -          | 53.73      |
> | Skyformer [5]            | 38.69      | 64.70     | 82.06       | 40.77      | 70.73       | -          | 59.39      |
> | cosFormer [6]            | 37.90      | 63.41     | 61.36       | 43.17      | 70.33       | -          | 55.23      |
> | FNet [7]                 | 35.33      | 65.11     | 59.61       | 38.67      | 77.80       | -          | 55.30      |
> | FLASH-quad [8]           | 42.20      | 64.10     | 83.00       | 48.30      | 63.28       | -          | 60.18      |
> | FLASH  [8]               | 38.70      | 64.10     | _86.10_     | 47.40      | 70.25       | -          | 61.31      |
> | TransNormer T1 [8]       | 41.03      | 66.90     | 83.11       | 51.60      | 75.92       | -          | 63.71      |
> | TransNormer T2  [8]      | 41.60      | 72.20     | 83.82       | 49.60      | 76.80       | -          | 64.80      |
> | KDEformer  [9]             | 36.64      | 62.00     | 73.52       | 45.45      | 68.13       | -          | 57.15      |
> | Hedgehog  [10]            | 37.15      | 64.60     | 82.24       | 40.15      | 74.16       | -          | 59.66      |
> | Transformers + Rotary [11] | _47.90_    | _79.08_   | 82.31       | **75.04**  | 76.64       | _84.72_    | _72.89_    |
> | DenseAttention  (ours)      | **50.50**  | **81.19** | **87.51**   | _72.55_    | **87.40**   | **88.82**  | **75.83**  |
>
> DenseAttention outperforms all other Transformer-based models on the Long Range Arena. The metrics for all tasks is accuracy. To ensure consistent comparisons, the averages for the models which report the result on Path-X task are computed without it. The references in the table link to the papers which reported LRA results for corresponding models.

---

> ### Author Response · Authors · 2024-11-27
> **General Response. Additional comparisons on the LRA (2)**
>
> **References**
>
> [1] Tay et al., "Long Range Arena: A Benchmark for Efficient Transformers." ICLR 2021
>
> [2] Peng et al., "Random Feature Attention." ICLR 2021
>
> [3] Ma et al., "Luna: Linear Unified Nested Attention." NeurIPS 2021
>
> [4] Xiong et al., "Nyströmformer: A Nyström-Based Algorithm for Approximating Self-Attention." AAAI 2021
>
> [5] Chen et al., "Skyformer: Remodel Self-Attention with Gaussian Kernel and Nyström Method." NeurIPS 2021
>
> [6] Qin et al., "cosFormer: Rethinking Softmax in Attention." ICLR 2022
>
> [7] Lee-Thorp et al., "FNet: Mixing Tokens with Fourier Transforms." NAACL 2022
>
> [8] Qin et al., "The Devil in Linear Transformer." EMNLP 2022
>
> [9] Zandieh et al., "KDEformer: Accelerating Transformers via Kernel Density Estimation." ICML 2023
>
> [10] Zhang et al., "The Hedgehog & the Porcupine: Expressive Linear Attentions with Softmax Mimicry." ICLR 2024
>
> [11] Amos et al., "Never Train from Scratch: Fair Comparison of Long-Sequence Models Requires Data-Driven Priors." ICLR 2024
>
> [12] Dao et al., "FlashAttention: Fast and Memory-Efficient Exact Attention with IO-Awareness." NeurIPS 2022

---

> ### Author Response · Authors · 2024-12-03
> **General Response. Part 3**
>
> **Conclusion and Future Work**
>
> In this paper, we propose DenseAttention Network -- a general architecture which simplifies the Transformer block and can serve as a drop-in replacement in every model architecture using it. We conduct experiments on the diverse modalities spanning from logic to language modeling and image classification and from short to extremely long sequence lengths using the LRA suite of benchmarks and MLM-style language model pre-training on text data. The results show that DenseAttention is capable of generalizing to many different tasks and context sizes and achieving favorable performance in comparison with standard Transformer and its augmented variants while being faster and more computationally efficient even with no specialized, low-level computation algorithms such as in [1].
>
> We acknowledge that there are other modalities and specialized architectures that would benefit from long-context efficiency improvements if the DenseAttention is ported or applied to them, such as ViT [2] and SAM [3] for Computer Vision tasks, and LLAMA
> [4] for decoder-style language modeling. We hope to address them in future work. In particular, we look forward to adapting DenseAttention architecture to causal LLAMA-style LLMs and studying their scaling laws at billions of parameters range.
>
> References:
>
> [1] Dao et al., "FlashAttention: Fast and Memory-Efficient Exact Attention with IO-Awareness." NeurIPS 2022
>
> [2] Dosovitskiy et al., "An Image is Worth 16x16 Words: Transformers for Image Recognition at Scale." ICLR 2021
>
> [3] Kirillov et al., "Segment Anything." ICCV 2023
>
> [4] Touvron et al., "LLaMA: Open and Efficient Foundation Language Models." arXiv preprint arXiv:2302.13971, 2023.
>
> ---
>
> **Other sub-quadratic algorithms for sequence processing**
>
> Another promising line of work focuses on applying deep State Space Models (SSMs) [1-4] and Linear RNNs [5-7] to long-range sequence and language modeling. Fundamentally, these architectures model interactions in sequence dimension by a linear recurrence:
>
> $
> \begin{align*}
> &x_t = \mathbf{\overline{A}}x_{t-1}  + \mathbf{\overline{B}}u_t, \\
> & y_t = \mathbf{\overline{C}}x_t + \mathbf{\overline{D}}u_t,
> \end{align*}
> $
>
> where recurrence matrix $\mathbf{\overline{A}}$ and other parameters are data-independent matrices which form and initialization are defining properties for a particular SSM/ RNN architecture.
> The linear recurrence is advantageous during inference as it runs in $O(N)$ time. For training, it also can be unrolled into a convolutional kernel $\mathbf{K} = \begin{bmatrix} \mathbf{\overline{C}  \overline{B}}, & \mathbf{\overline{C} \overline{A} \overline{B}}, & \ldots, & \mathbf{\overline{C} \overline{A}}^{N-1}\mathbf{\overline{B}} \end{bmatrix} $ to compute $y = \mathbf{K} * u$ via Fast Fourier Transform (FFT) in $O (N \log N)$ time. Here, we set $\mathbf{D}=0$ for ease of exposition, but in practice it's usually set to identity to act as a skip-connection ubiquitous in modern deep NN architectures.
>
> Among other novel algorithms which rely on FFT or its generalizations such as Monarch matrices [8], are Long Convolutions [9], Hyena [10], and Monarch Mixer [11], with the latter using sub-quadratic primitives both for computations along the sequence length and the model dimension.
>
> While being sub-quadratic, these algorithms are still slower than linear time as in DenseAttention. However, recently [4] introduced data-dependent gating for SSM parameters and low-level, hardware efficient CUDA implementation for parallel-scan operation which allows for fast linear-time processing both during training and inference. And [12] adopt a similar gating mechanism for causal linear attention which allows to drop the denominator in the linear attention formula but also admits no parallel training without resorting to low-level implementations.
>
> References:
>
> [1] Gu et al., "Efficiently Modeling Long Sequences with Structured State Spaces." ICLR 2022
>
> [2] Gupta et al., "Diagonal State Spaces are as Effective as Structured State Spaces." NeurIPS 2022
>
> [3] Ma et al., "Mega: Moving Average Equipped Gated Attention." ICLR 2023
>
> [4] Gu and Dao, "Mamba: Linear-Time Sequence Modeling with Selective State Spaces." CoLM 2024
>
> [5] Beck et al., "xLSTM: Extended Long Short-Term Memory." NeurIPS 2024
>
> [6] Orvieto et al., "Resurrecting Recurrent Neural Networks for Long Sequences." ICML 2023
>
> [7] Peng et al., "RWKV: Reinventing RNNs for the Transformer Era." Findings of EMNLP 2023
>
> [8] Dao et al., "Monarch: Expressive Structured Matrices for Efficient and Accurate Training." ICML 2022
>
> [9] Fu et al., "Simple Hardware-Efficient Long Convolutions for Sequence Modeling." ICML 2023
>
> [10] Poli et al., "Hyena Hierarchy: Towards Larger Convolutional Language Models." ICML 2023
>
> [11] Fu et al., "Monarch Mixer: A Simple Sub-Quadratic GEMM-Based Architecture." NeurIPS 2023
>
> [12] Yang et al., "Gated Linear Attention Transformers with Hardware-Efficient Training." ICML 2024

---

> ### Author Response · Authors · 2024-12-04
> **General Response. Part 4. Additional experiments and ablations.**
>
> **New results of DANet-BERT pre-training**
>
> | **Model**                            | **MLM Loss (L=128)** | **Acc. (L=128)** | **MLM Loss (L=512)** | **Acc. (L=512)** | **MLM Loss (L=1024)** | **Acc. (L=1024)** |
> |--------------------------------------|-----------------------|-------------------|-----------------------|-------------------|-----------------------|-------------------|
> | DenseAttention (1 head, N=128)       | **1.91**             | **0.620**         | -                    | -                 | -                     | -                 |
> | DenseAttention (4 heads, N=128)      | 1.97                 | 0.611             | -                    | -                 | -                     | -                 |
> | BERT-large                           | 2.58                 | 0.582             | 2.31                 | 0.614              | -                     | -                 |
> | DenseAttention (1 head, N=512)       | 1.97                 | 0.614             | 1.726                 | 0.644             | -                     | -                 |
> | DenseAttention (4 heads, N=512)      | 2.04                 | 0.602             | 1.84                 | 0.624             | -                     | -                 |
> | DenseAttention (1 head, N=1024)      | 1.96                 | 0.615             | **1.71**             | **0.648**         | 2.26                  | 0.591             |
> | DenseAttention (4 heads, N=1024)     | 2.07                  | 0.598             | 1.83                 | 0.627             | **1.87**             | **0.618**         |
>
> Evaluations of MLM loss and accuracy for DenseAttention Network models and the original BERT on C4 dataset texts of different context sizes. N is the maximum sequence length with which a model was trained or evaluated and L is the length of evaluation samples. DANet-BERT model variations uniformly outperform the original BERT on corresponding context sizes.
>
> The results of the new DANet-BERT model, which is closely aligned with the original, are similar with the old results. The relations and trends in performance between all models and all context lengths remain unchanged.
>
> ---
>
>
> **DANet-BERT 16K with local attention**
>
> We conducted additional experiments by taking DANet-BERT model after it had finished pre-training on sequence length 512 and continuing pre-training on sequence lengths 1024 and then 16384 tokens both with and without local attention scheme.
>
> | **Context size**      | **1k**                    |                           |                           | **16k**                    |                           |                           |
> |------------------------|---------------------------|---------------------------|---------------------------|----------------------------|---------------------------|---------------------------|
> | **Metrics**           | **Samples**              | **MLM loss**              | **MLM acc.**              | **Samples**               | **MLM loss**              | **MLM acc.**              |
> | DANet-BERT         | 80M                      | 2.255                     | 0.591                     | 27M                       | 2.843                     | 0.452                     |
> | DANet-BERT + local attention | 80M            | 1.705                     | 0.647                     | 7.8M                      | 1.689                     | 0.637                     |
>
> Comparison of DenseAttention BERT-large pre-trained on long context sizes with and without local attention. The models with context size 1k and 16k were evaluated on the corresponding length texts from C4 and Bookcorpus (held-out split) datasets respectively. Samples denotes number of sequences of corresponding length seen by a model during continual pre-training.
>
> The results show that introduction of local-global attention pattern helps to quickly recover the modeling performance even on extremely long sequences. It brings the performance to the same level we observed when pre-training on small sequences and significantly outperforms the models which were pre-trained without the local attention.
>
> ---
>
> **Ablation on use of MaxNormActivation in standard Transformer**
>
> | **Model** | **MLM loss** | **MLM accuracy** |
> |-------------------------------------|--------------|-------------------|
> | BERT-large (LayerNorm) | 2.11 | 59.3 |
> | BERT-large (MaxNormActivation) | 2.46 | 54.3 |
>
> Comparisons between LayerNorm and MaxNormActivation for BERT-large Transformer pre-trained on Wiki+BookCorpus dataset for 10B tokens. MLM loss and accuracy are reported for out-of-sample data from C4 dataset
>
> The results indicate that standard LayerNorm is optimal for standard Transformer as replacing it by MaxNormActivation leads to a subpar performance. On the other hand, MaxNormActivation is not a merely optimal but rather essential part of DANet architecture, because putting the standard LayerNorm into it instead of MaxNormActivation results in numerical instability.

---

### Author Response · Authors · 2024-12-04
**General summary**

We sincerely thank reviewers for their time, feedback, and constructive suggestions which led to additional experiments and helped to clarify and strengthen our work. We would like to emphasize the main points of the paper, highlighted strengths, and key updates to the revision.

---

**Recap**

In this paper we propose DenseAttention Network, a simplification of Self-Attention and the vanilla Transformer, which runs both in $O(N)$ and $O(N^2)$ time depending on what’s more computationally efficient.  Among other modifications to the architecture, we completely eliminate softmax from attention with no substitutions to it. We show it’s able to work if the standard LayerNorm is replaced with a proposed MaxNormActivation. DenseAttention in both regimes runs faster than highly-optimized low-level Transformer implementations, despite being coded in plain PyTorch. It retains or outperforms the modeling quality of the Transformer and its multiple variations across the wide range of context sizes, as shown in experiments on language modeling.

Additionally, we suggest two extensions, Cosine RelPE as a computationally efficient alternative to RoPE, and local-global attention scheme, to aid the model performance on extremely long sequences while maintaining computational efficiency. It allows us to achieve state-of-the-art results on the Long Range Arena suite of benchmarks among all Transformer-based models and the overall best result on the Pathfinder-256 task (65K context size).

We are grateful to reviewers for acknowledging the following strengths in our work:

* The paper being **clearly written and easy to follow** as well as proposing a solution to **important** problem of **efficiency** of Transformer architectures (ruBJ).

* The solution being  **well-motivated** (ruBJ, oGkz), in particular, MaxNormActivation, which is also acknowledged to be **useful** primitive **to stabilize LLM training** (rVwA).

* Originality, novelty and efficiency of the approach to unite and **fuse** projection matrices of Transformer into **single parameter** (HAtu, rVwA) and of the proposition to **use local attention** together with the new architecture (HAtu).

* Strong empirical results for speed/ **efficiency** gains with just **architectural** changes and **no specialized kernels** as well as for promising modeling performance in **long-context sequence modeling tasks** (rVwA).

* **Provision of code** (oGkz) which ensures easy reproducibility.

**Summary of updates**

Here’s the summary of the key updates we made based on reviewers’ comments and suggestions.

* Added a discussion of differences of Linear Transformers in relation to DenseAttention in **Appendix D** of the revised manuscript (also in **General Response. Part 1** for convenience) (ruBJ, HAtu, oGkz, rVwA). Additionally provided an exposition of other sub-quadratic algorithms for sequence processing in **General Response. Part 3** (please note: version of this exposition in the manuscript is a draft which will be updated). (oGkz).

* Added an “Extended comparison with Transformer-based models” on the LRA for 25+ models in **Appendix E.2** (also in **General Response. Part 2**), which highlights strong performance of DANet (ruBJ, HAtu, rVwA). Provided an extended “Discussion of the LRA tasks” clarifying the scope and difficulty for each task in **Appendix E.1** (ruBJ, oGkz).

* Conducted additional experimental studies on extremely long sequences performance on Pathfinder-256 benchmark in **Appendix H.1** (ruBJ, oGkz), and scaling effects in  **Appendix H.3** (ruBJ, rVwA), both with positive outcomes.

* Reproduced all DANet-BERT pretraining experiments with architecture equivalent to original BERT in **General Response. Part 4** and arrived at results and conclusions similar to former ones (oGkz).

* Performed additional ablation studies on local-global attention in DANet-BERT for extremely long context sizes in **General Response. Part 4** (oGkz), speed gains of Cosine PE in **Appendix H.2** (oGkz, rVwA), and use of MaxNormActivation in vanilla Transformer in **General Response. Part 4** (rVwA), all to expected/ favorable outcomes.

* Added a “Conclusion and Future Work” section in **Appendix A** (draft) and **General Response. Part 3** (final version) to clarify the scope of the work and future directions (ruBJ, oGkz).

---

### Meta-Review · Area_Chair_bjZq · 2024-12-18

**Metareview:**

The primary goal of this paper is to develop an attention mechanism which, given an input of N tokens, allows for the computation of the full matrix of NxN pairwise attention weights in less than N^2 time (i.e., linear in N).  This is done in a manner similar to the Linear Attention mechanism, but as discussed by the reviews and authors is still somewhat distinct from Linear Attention in a few technical details regarding issues like normalization, which the authors suggest is important for numerical stability.  Beyond this, the authors also make several changes to the overall architecture in terms of other normalization operators, positional encoding, etc and achieve good performance on several benchmarks.

Overall, the reviewers are fairly unanimous in leaning to rejection, and I agree that the paper is perhaps not ready for publication just yet.  As has been noted in the reviews, the proposed mechanism is very similar to Linear Attention, and while the current work may not be strictly captured by Linear Attention, this proximity combined with the numerous other changes to the architecture make it difficult to establish which elements are critical to performance and for readers to understand the key aspects of the contribution and why the proposed mechanism is needed relative to Linear Attention.  I would encourage the authors to take these aspects and additional comments from the reviewers into account when preparing a new manuscript for a future conference which better highlights the advantages of the proposed approach relative to existing techniques (such as Linear Attention) as well as justifying the additional modification to the overall architecture.

**Additional Comments On Reviewer Discussion:**

The authors have provided very extensive rebuttals to the initial reviews, and some of the reviewers found these arguments helpful in showing the contribution of the work and revised their scores higher (though not to the point of arguing for paper acceptance).  While this is a commendable effort from the authors, this also somewhat suggests that the manuscript also would potentially benefit from a significant revision to incorporate the clarification and additional content from the rebuttal, and as mentioned in the meta review I encourage the authors to take this into account and look forward to their submission at future conferences.

---

### Decision · Program_Chairs · 2025-01-22

Reject